# Platelets favor the outgrowth of established metastases

Maria J. Garcia-Leon[1,2,3,4,10,12] ✉, Cristina Liboni[1,2,3,4,12],
Vincent Mittelheisser[1,2,3,4,12], Louis Bochler[1,2,3,4], Gautier Follain[1,2,3,4,11],
Clarisse Mouriaux[5], Ignacio Busnelli[1,2,3,4], Annabel Larnicol[1,2,3,4],
Florent Colin[1,2,3,4], Marina Peralta[1,2,3,4], Naël Osmani[1,2,3,4],
Valentin Gensbittel[1,2,3,4], Catherine Bourdon[5], Rafael Samaniego[6],
Angélique Pichot[2,3,7], Nicodème Paul[2,3,7], Anne Molitor[2,3,7],
Raphaël Carapito[2,3,7,8], Martine Jandrot-Perrus[9], Olivier Lefebvre[1,2,3,4],
Pierre H. Mangin[5] ✉ & Jacky G. Goetz[1,2,3,4] ✉

Despite abundant evidence demonstrating that platelets foster metastasis, anti-platelet agents have low therapeutic potential due to the risk of hemorrhages. In addition, whether platelets can regulate metastasis at the late stages of the disease remains unknown. In this study, we subject syngeneic models of metastasis to various thrombocytopenic regimes to show that platelets provide a biphasic contribution to metastasis. While potent intravascular binding of platelets to tumor cells efficiently promotes metastasis, platelets further support the outgrowth of established metastases via immune suppression. Genetic depletion and pharmacological targeting of the glycoprotein VI (GPVI) platelet-specific receptor in humanized mouse models efficiently reduce the growth of established metastases, independently of active platelet binding to tumor cells in the bloodstream. Our study demonstrates therapeutic efficacy when targeting animals bearing growing metastases. It further identifies GPVI as a molecular target whose inhibition can impair metastasis without inducing collateral hemostatic perturbations.

Metastasis is a pathological process during which tumor cells (TCs) use the bloodstream to colonize distant organs. Yet, the fundamental mechanisms driving metastasis remain poorly understood, limiting the development of efficient therapeutic schemes. When disseminating, circulating TCs (CTCs) face several hostile forces that are detrimental to their survival[1]. Yet, they can exploit hemodynamics to arrest and successfully grow upon extravasation[2–4]. In addition, CTCs efficiently partner with the

[1]Tumor Biomechanics, INSERM UMR_S1109, Strasbourg, France. [2]Université de Strasbourg, Strasbourg, France. [3]Fédération de Médecine Translationnelle de Strasbourg (FMTS), Strasbourg, France. [4]Equipe Labellisée Ligue Contre le Cancer, Paris, France. [5]UMR_S 1255, INSERM, Etablissement Français du Sang-Alsace, Université de Strasbourg, F-67000 Strasbourg, France. [6]Instituto de Investigación Sanitaria Gregorio Marañón (IISGM), Unidad de Microscopía Confocal, Madrid, Spain. [7]Laboratoire d'ImmunoRhumatologie Moléculaire, Plateforme GENOMAX, Institut national de la santé et de la recherche médicale (INSERM) UMR_S 1109, Institut thématique interdisciplinaire (ITI) de Médecine de Précision de Strasbourg Transplantex NG, Faculté de Médecine, France. [8]Service d'Immunologie Biologique, Plateau Technique de Biologie, Pôle de Biologie, Nouvel Hôpital Civil, Hôpitaux Universitaires de Strasbourg, 1 Place de l'Hôpital, 67091 Strasbourg, France. [9]UMR_S 1148, INSERM, Université Paris-Cité, Paris, France. [10]Present address: Domain therapeutics, Parc d'Innovation – 220 Boulevard Gonthier D'Andernach, 67400 Strasbourg – Illkirch, France. [11]Present address: Turku Bioscience Centre, University of Turku and Åbo Akademi University, Turku, Finland. [12]These authors contributed equally: Maria J. Garcia-Leon, Cristina Liboni, Vincent Mittelheisser. ✉ e-mail: mgarcialeon@domaintherapeutics.com; Pierre.Mangin@efs.sante.fr; jacky.goetz@inserm.fr

bloodstream ecosystem and interact with blood components on their way to metastasis[5–8].

Among these, platelets are perfect allies: they promote metastatic progression[9] by protecting CTCs from destruction by shear forces, mediate immune evasion, favor adhesion and extravasation at distant sites, facilitate neo-angiogenesis, and sustain a tumor-prone inflammation[10]. Although recent work suggests that platelets can also hamper tumor initiation and growth, as shown in hepatic[11–13] and ovarian[11] tumors, they have mostly been shown to promote TC metastasis. A seminal study documented the inhibition of metastasis by platelet depletion, which can be rescued by platelets transfusion[14]. The depletion of platelet receptors, such as the vWF receptor GPIb, and the collagen receptor GPVI, results in the inhibition of lung metastasis of melanoma and lung carcinoma cells, respectively[15–17].

Anti-platelet strategies became inevitably obvious candidates for impairing metastasis. First, in the bloodstream, CTCs rapidly bind, activate, and aggregate circulating platelets. This provides a physical shield to CTCs and favors intravascular arrest and survival[18,19], as well as successful extravasation and metastatic outgrowth[20–22]. While it is thus tempting to target the intravascular CTCs-platelet interaction with anti-platelet therapies for impairing metastasis, whether this is a universal feature of cancers remains unexplored. It is thus likely that platelets' contributions may differ depending on cancer types[23,24], leading to an unclear outcome when using anti-platelet drugs in oncologic patients[25], in addition to the fact that they cause bleeding. Second, our current knowledge of the pro-metastatic function of platelets focuses on their contribution to the intravascular behavior of CTCs, which subsequently impacts metastatic fitness[10,20,21]. Whether platelets are capable of tuning extravascular and late metastatic steps remain unknown and unexplored. Yet, they could offer additional means for targeting platelets during metastasis. Finally, the development of efficient anti-platelet strategies to fight metastasis faces major pitfalls inherent to the properties of platelets[26,27]. The majority the current of anti-platelet strategies are accompanied by increased bleeding risk[28] and fail when used against metastasis. As this remains a major concern, the use of anti-platelet therapies remains unexplored in the clinic.

Building on such limitations, we designed a study aiming at interrogating which steps of the metastasis cascade are controlled by platelets and whether this is impacted by platelet binding. We first demonstrated that such binding is heterogenous among metastatic TCs. We selected two mouse TCs lines and subjected them to several thrombocytopenic (TCP) regimes in syngeneic mouse models to interrogate the timings at which platelets mostly contribute to metastatic outgrowth. We show that platelets are crucial for the adhesion, clustering, and survival of TCs within the vasculature and we provide further evidence that platelets support their long-term metastatic potential in vivo. We observed that platelets efficiently colonize metastatic foci, in both murine and human samples. Depletion of platelets in animals carrying growing metastases was sufficient to reduce metastatic outgrowth, demonstrating that platelets favor metastasis independently of platelet-TC interaction in the bloodstream and mostly via immune suppression, which could be targeted with new treatment strategies. We focused our attention on the platelet receptor GPVI, which is exclusively expressed at the surface of platelets and whose targeting inhibits platelet function with no impact on bleeding. Genetic depletion and pharmacological targeting of the platelet-specific receptor GPVI in humanized mouse models efficiently reduced the growth of established metastases, independently of active platelet binding to TCs in the bloodstream. Our findings highlight, distinctly from previous studies, that interfering with GPVI can impair metastatic outgrowth of already-established metastases. We believe that such discovery paves the way for establishing successful therapeutic intervention in patients where the primary tumor has already generated one or several metastatic sites.

## Results

### TCs bind and activate platelets in vitro with different efficiencies

Platelets have been reported to bind CTCs[9] but whether this applies to every cell type remains unknown. We first checked in vitro the efficiency of human platelets to interact with various human and mouse TC lines, either from breast cancer (epithelium-derived) or melanoma, using scanning electron microscopy (SEM). We found out that platelets do not equally bind TCs (Fig. 1A, B). While some cells, such as 4T1 or MCF7, efficiently bind platelets, others, such as B16F10 or MDA-MB-231, cannot bind more platelets than passive beads (Fig. 1A, B and supplementary Fig. 1A). As platelets' shape provides information on their potential to be activated by TCs, we assessed their morphology (Fig. 1C, D) and observed that TCs behave as weak platelet agonists. We probed their activation through their ability to promote tumor-induced platelets activation (TCIPA) (Fig. 1E, F), which correlated with platelet binding. We further observed that TCs fail to promote aggregation (supplementary Fig. 1B, D) although they bind platelets also in the absence of plasma proteins (supplementary Fig. 1E, F) or activate them (supplementary Fig. 1G, H), altogether indicating that TCs do not equally bind and activate platelets, questioning how and when platelets impact the metastatic properties of TCs. To do so, we selected two mouse TC lines known for their high metastatic properties (4T1 vs B16F10) and that are compatible with experimental metastasis approaches in a syngeneic background where platelet counts can be easily manipulated.

### Platelets may orchestrate the intravascular behavior of CTCs and may control lung seeding in vivo

It is well-established that the early seeding of TCs shapes their metastatic potential[29]. Thus, we next interrogated whether differential platelet binding may influence metastasis in vivo. We designed an experimental lung metastasis assay that allows us to probe the behavior of TCs at very early and sequential steps, including late ones, of the metastatic cascade (Fig. 2A). We depleted platelets using an α-GPIb treatment that proved to be equivalently efficient in both animal models (BALB/c and C57BL6/J, Fig. 2B) and assessed TCs initial lung seeding by sequential in vivo imaging of 4T1 or B16F10 luciferase-expressing cells, starting from 15 minutes post injection (mpi) up to 24 hours post injection (hpi) (Fig. 2A). 4T1 cells reached and efficiently seeded control lungs from 15mpi to 24hpi (33-fold), as shown previously for metastatic cells[29]. However, when 4T1 cells were injected in TCP mice, lung seeding was drastically reduced by 10-fold when compared to control conditions. In contrast, the lung seeding potential of B16F10 cells was independent of the presence of platelets (Fig. 2C, D). These data suggest that lung seeding of known metastatic TCs does not necessarily depend on platelets and their platelet binding potential.

To provide a live and quantitative description of this platelet-dependent intravascular arrest of TCs, we next used our previously established experimental metastasis model in the zebrafish embryo[2]. Fluorescently-labeled TCs and human citrated platelet-rich plasma (cPRP) were co-injected in the duct of Cuvier of *Tg(Fli1a:GFP)* embryos at 2 days post-fertilization (dpf) and live-monitored during 4 mpi (Fig. 2E). Careful live tracking of TCs[2] showed that platelets favor the stable arrest of 4T1 in the caudal plexus (Fig. 2F, supplementary Fig. 2A) thus reducing their circulation time (supplementary Fig. 2A). In contrast, platelets did not affect the circulation and arrest patterns of B16F10 (Fig. 2G, supplementary Fig. 2B). A heatmapping analysis, allowing quantitative location of the hotspot of arrested TCs, revealed that platelets favored the stable arrest of 4T1 cells in high shear arterial vessels (DA, dorsal aorta; CV, caudal vein) without impacting the entry of TCs into small sized vessels (ISV, intersegmental vessels, supplementary Fig. 2A, C, D) where arrest is mostly imposed by physical constraints. Interestingly, although platelets did not affect the number of B16F10 stable arrest events in non-constraining arterial vessels, they

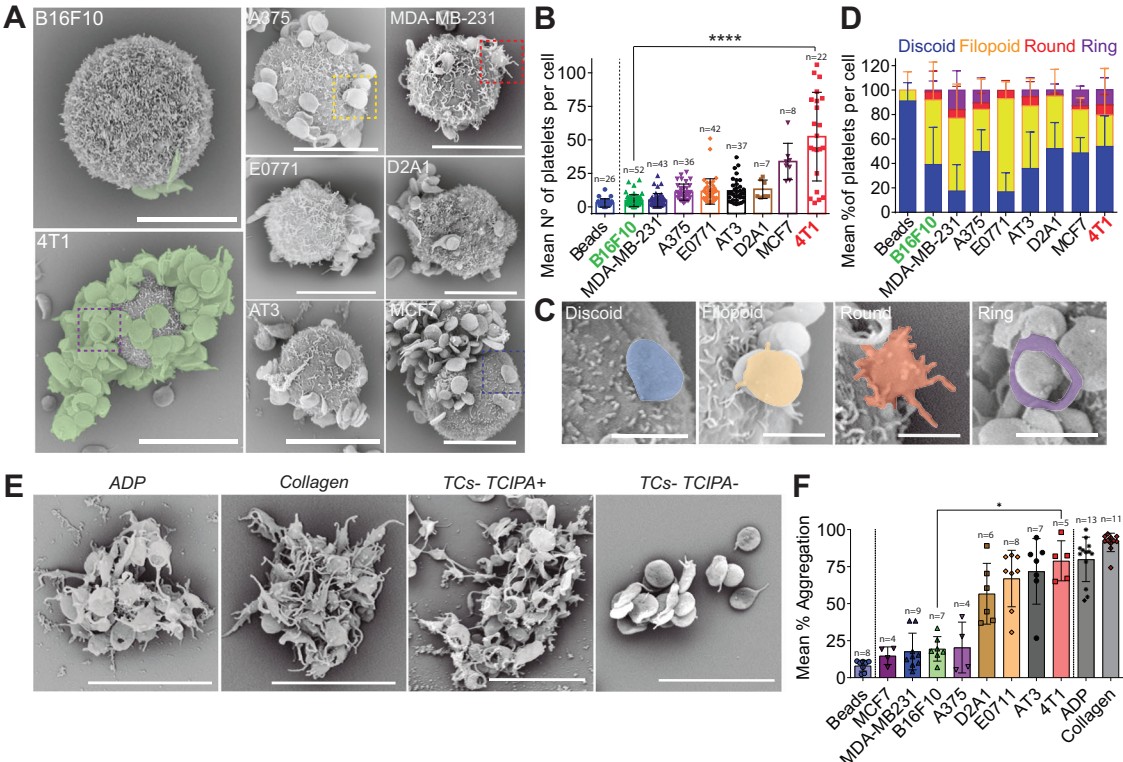

**Fig. 1 | TCs bind and aggregate blood platelets with different efficiencies. A** SEM images of TCs interacting with mouse platelets contained within mouse citrated platelet-rich plasma (cPRP). Mouse breast carcinoma (4T1) and mouse melanoma (B16F10) cells are highlighted. Scale bar: 10 µm. **B** Platelet binding profiles quantification graph of **A**. Number of platelets per cell for *n* cells (the number of cells per cell line analyzed are displayed above the histogram) are shown as mean ± SD and analyzed using Kruskal–Wallis with False discovery rate post-test (1 experiment was performed for MDA-MB-231, E0771 and D2A1, 2 for MCF7 and A375, and 3 for AT3 and B16F10), *q* < 0.0001. **C** SEM images of TC-bound platelets at different activation states including discoid (resting), filopodia (low activation), round (full activation) and ring-shaped platelets (unknown function). Scale bar: 2 µm. **D** Platelet shape profiles quantification graph of **C**. Relative amount platelet shape per TCs for *n* cells or beads (beads *n* = 22, B16F10 *n* = 48, MDA-MB-231 *n* = 38, A375 *n* = 36, E0771 *n* = 42, AT3 *n* = 37, D2A1 *n* = 8, MCF7 *n* = 8, 4T1 *n* = 21) are shown as mean ± SD (2 and 6 independent experiments were performed for 4T1 and B16F10, respectively); activation stages are color-coded. **E** SEM images of mouse platelet aggregates after platelet aggregation assay using cPRP with ADP and collagen as classic agonists, and 4T1 and B16F10 TCs TCIPA+ or TCIPA− cells. Scale bar: 8 µm. **F** Platelet aggregation quantification graph of **E**. Relative human cPRP platelet aggregation for *n* aggregation curves (the number of aggregation curves per cell line analyzed are displayed above the histogram) are shown as mean ± SD and analyzed using Kruskal–Wallis with False discovery rate post-test, four independent aggregation experiments were performed, *q* = 0.014. Source data are provided as a Source Data file.

favored their entry and arrest into ISV (supplementary Fig. 2B–D). We next probed arrest events at the nanoscale using correlative light electron microscopy (CLEM)[2,30] and observed that stably arrested 4T1 cells efficiently interacted with platelets with an activated morphology (filopodia, Fig. 2H) and endothelial cells. Altogether, this provides the innovative in vivo demonstration that platelet binding favors the intravascular arrest and adhesion of CTCs to the endothelial wall, which can be probed at nanoscale (CLEM).

We next analyzed TC-platelet binding in mouse lung sections containing early-seeded (15mpi and 2hpi) 4T1 or B16F10 TCs (supplementary Fig. 3A). While imaging at 15mpi allows catching the very initial in vivo behavior of TCs, around 50% of the input cells are already cleared from the lungs after 2hpi in the absence of platelets (Fig. 2C, D), thus identifying a critical platelet-dependent step. 3D high-resolution imaging at 15mpi revealed that 4T1 cells significantly bind, recruit and aggregate platelets in vivo (Fig. 2I), validating our initial in vitro observations. As expected, the size (volume) and number of platelet aggregates were also significantly lower in TCP animals (supplementary Fig. 3B). While they are initially low as observed in vitro, the number of platelets found near B16F10 cells increased significantly at 2hpi (Fig. 2J) suggesting that platelets are passively recruited in vivo to arrest sites, either by blood flow clogging, TCs cytoplasmic release of platelet aggregation inducers, or other mechanisms. When probing cell survival using cell morphometrics at 2hpi, we observed increased nuclear fragmented (non-viable) 4T1 cells in α-GPIb-treated animals

(supplementary Fig. 3A, C, D) suggesting that platelets favor intra-vascular survival of 4T1 cells, as previously reported[18]. Interestingly, nuclear fragmentation levels of B16F10 were unperturbed upon thrombocytopenia. Because platelets can passively be recruited at TCs 2hpi (supplementary Fig. 3E) upon flow clogging, they are likely to favor the intravascular survival of neighboring viable arrested cells, as previously observed[31]. We further probed the ability of platelets to promote the clustering of CTCs, which is known to favor seeding and metastatic potential[5]. In vitro, platelets favor the clustering of 4T1 cells but not B16F10 cells (supplementary Fig. 3F, G), in line with their ability to bind platelets (supplementary Fig. 3G and Fig. 1A, B). Of note, 4T1 cells form more and bigger clusters than B16F10 cells, whether alone or when bound to platelets (supplementary Fig. 3G, H), with a direct impact on cell survival (supplementary Fig. 3I) as observed in vivo (supplementary Fig. 3E). In conclusion, using a novel in vivo analysis of early lung metastatic seeding in mice and zebrafish embryos, we show that platelet counts do not equally impact the lung seeding of metastatic TCs although they strongly orchestrate their clustering potential and early intravascular survival.

## Platelets control metastatic fitness at early stages

We next wondered whether a positive impact of platelets on lung seeding would translate into efficient metastatic outgrowth. We longitudinally tracked metastatic burden over time in control and TCP mice after a single treatment with α-GPIb antibody prior TCs injection

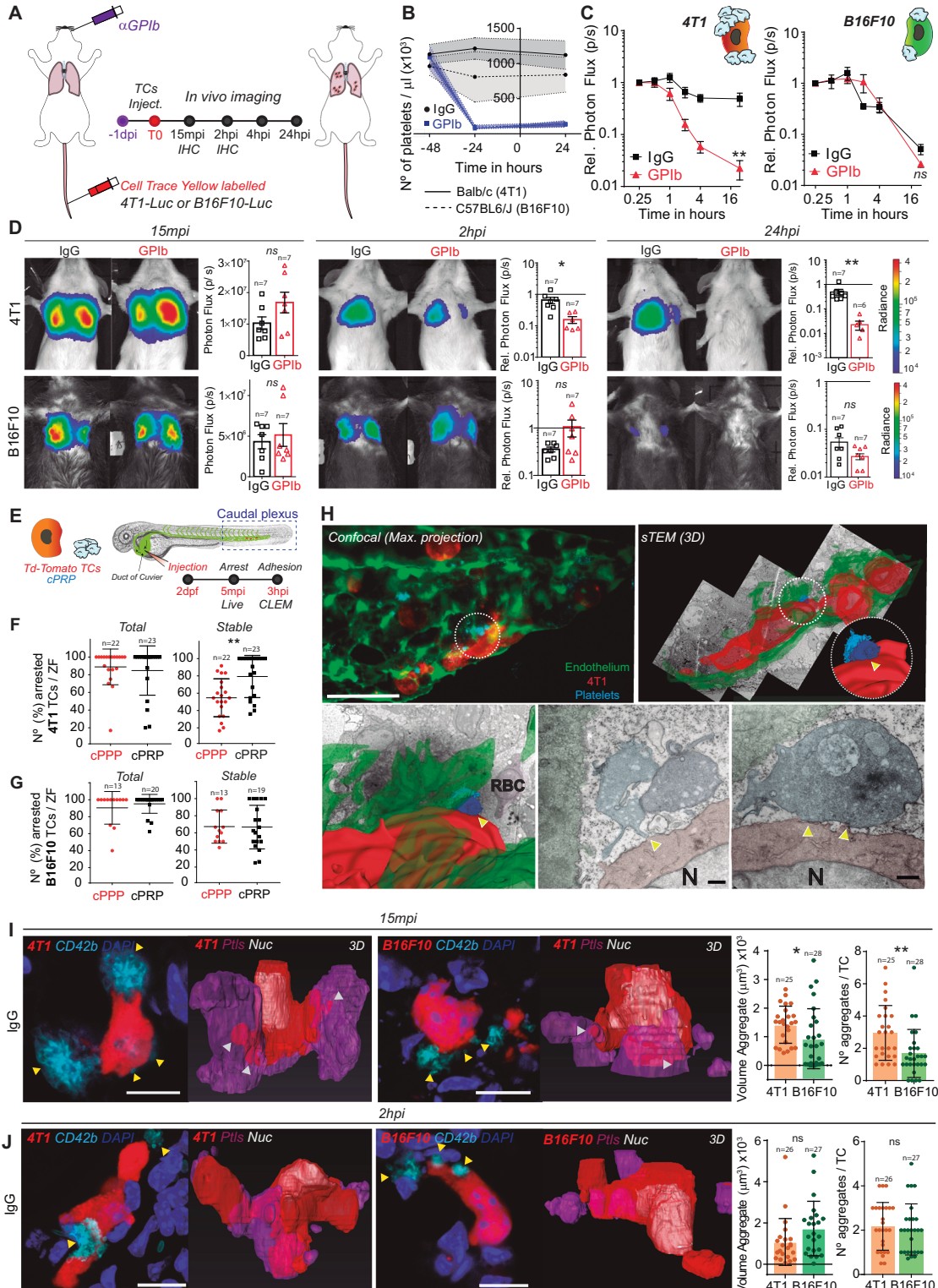

(14 days, Fig. 3A). This model, where platelet counts quickly recovered and reached approximately 50% of the normal numbers at day 7 (Fig. 3B), was referred as short-term thrombocytopenia (shortTCP).

Metastatic outgrowth (day 14) was significantly reduced by shortTCP (Fig. 3C). Interestingly, shortTCP impacted the growth of B16F10 metastases only after lung seeding. Indeed, although B16F10 lung seeding was equivalent at 24hpi (Figs. 2C, D, 3C, D), the metastatic

burden was significantly reduced by shortTCP (14dpi, Fig. 3C, D). While we cannot exclude that metastatic outgrowth is impacted by the quick recovery of the platelet counts (Fig. 3B), these data demonstrate that platelets are likely to control initial events of metastatic seeding (day 1 post injection) that later translate into a significant reduction of metastatic outgrowth (day 14). Yet, these data also hint toward a secondary effect of platelets at later stages of metastasis, as observed in

**Fig. 2 | In vivo lung seeding of 4T1 or B16F10 TCs is differentially influenced by platelets. A** Infographics showing the experimental setting. **B** Graph showing platelet counts on IgG- and GPIb-treated BALB/c and C57BL6/J animals, as mean ± SD ($n = 7$ mice per group, 2 independent experiments were performed). **C** Relative bioluminescence kinetics of 4T1-Luc and B16F10-RedLuc TCs lung seeding during 24 hours in normal and TCP mice ($n = 7$ mice per group) are shown as mean ± SEM and analyzed using two-way ANOVA with False discovery rate post-test (2 independent experiments were performed), $q = 0.0028$. **D** Representative images and timepoint comparisons of relative lung bioluminescence in control versus TCP animals in 4T1 (up) and B16F10 (down) TCs models. Lung bioluminescence quantifications of $n$ mice (the number of mice analyzed is displayed above the histogram) are shown as mean ± SEM and analyzed using two-sided Mann–Whitney test (two independent experiments were performed), 4T1 2hpi $p = 0.0047$, 24hpi $p = 0.0012$. **E** Infographics showing the experimental setting to study the early steps of the metastatic cascade in vivo in the ZF embryo. **F, G** Quantification of total and stable arrest of 4T1 (**F**) and B16F10 (**G**) cells in the caudal plexus of the ZF embryo

during 5 minutes of live video microscopy. Arrested TCs of $n$ ZF embryos (the number of ZF embryos analyzed are displayed above the dot plots) are shown as mean ± SD and analyzed using two-sided Mann–Whitney test (4 independent experiments were performed), $p = 0.0034$. **H** Representative confocal and CLEM images of 4T1 cells arrested intravascularly in the caudal plexus of the ZF embryo after 3hpi. Arrowheads indicate contact between platelets and TCs. Scale: 50 μm (confocal); 500 nm (CLEM). **I, J** Representative confocal images of single 4T1 or B16F10 cells arrested in mouse lungs at 15mpi (**I**) and 2hpi (**J**). Arrowheads show the interaction of single platelets and platelet aggregates with the TCs. Scale bar: 10 μm. On the right, the volume and number of platelet aggregates around the arrested TCs is shown. Platelet and TCs volumes were calculated upon segmentation of TCs ($n$ values are displayed above the histograms) and platelets using the AMIRA software and are shown as mean ± SD and analyzed using two-sided Mann–Whitney test (1 experiment with 3 mice was performed for each timepoint), (**I**) $p = 0.034$ and $p = 0.002$. Source data are provided as a Source Data file.

the B16F10 model, where thrombocytopenia significantly diminishes metastatic burden at day 14 with, however, no impact on lung seeding (day 1).

To explore the mechanisms that may underpin this effect occurring after seeding, metastatic lungs were surgically resected on day 14. Macroscopic inspection of B16F10 metastatic foci revealed, as expected, that shortTCP significantly reduced their numbers (Fig. 3E, supplementary Fig. 4A), in agreement with the lower bioluminescent signal observed at day 14 (Fig. 3C, D). While the size of the B16F10 foci appeared similar (Fig. 3E), these lesions displayed a significant decrease in Ki67-positive cells, indicating that their proliferation index was impacted by platelet depletion (Ki67, Fig. 3G, H), although the latter had no effect on seeding. Interestingly, such an effect was not observed in 4T1 foci (Ki67, Fig. 3F, H, supplementary Fig. 4A), suggesting that platelets may tune the proliferation of B16F10 during the growth of already-established metastatic foci. While platelets were shown to promote the recruitment of a pro-inflammatory microenvironment favoring the proliferation of TCs[32], shortTCP did not affect immune cell recruitment in any of the two cell models (Fig. 3I–K). These results further demonstrate the versatility of the prometastatic effects of platelets and their ability to support metastatic outgrowth at timings (early seeding vs late outgrowth) that might differ depending on the TCs. Platelets were proven to colonize and support primary tumors[11,32,33]. Yet, whether they also populate and support metastatic foci remains to be demonstrated. We assessed the number of single (≤3 platelets) or aggregated (>3 platelets) intrametastatic platelets and observed that total platelet numbers and aggregates were diminished upon shortTCP in both models (supplementary Fig. 4B), despite the recovery in platelet counts (Fig. 3B). Importantly, while platelet numbers within control 4T1 and B16F10 metastases were equivalent (supplementary Fig. 4B), the number of aggregates within B16F10 metastases was significantly higher than in 4T1 metastases, hinting toward the idea that B16F10 cells are likely to better recruit, activate and aggregate platelets within metastatic foci (supplementary Fig. 4B, C). Altogether, these results demonstrate that while transient thrombocytopenia does not equally impact seeding over two cancer models (Fig. 2), it significantly perturbs their subsequent outgrowth. Because platelet aggregates populate metastatic foci and shape their proliferation index, it is tempting to speculate that they can shape late stages of metastasis.

## Platelets control late steps of metastatic outgrowth

Since platelets populate metastatic foci (supplementary Fig. 4B, C), where, as demonstrated, they are able to release important prometastatic molecules[34,35] and to circumvent the limitations of the yet classically used shortTCP model, we designed a long-term thrombocytopenia model (longTCP) by the sequential injection of α-GPIb mAb every three to four days, starting 24 h before TCs injection (Fig. 4A). Doing so, we could maintain platelet counts low up to day 10 post

injection (Fig. 4B) and therefore fully impair platelets interactions during the initial growth phase of micro-metastatic foci. A metastatic outgrowth of 4T1 cells was similarly reduced in longTCP (Fig. 4C, D) compared to shortTCP (Fig. 3C, D), confirming that platelets can strongly contribute to the early steps of the metastasis cascade. We further confirmed that such a platelet-dependent effect was not limited to lung metastasis and performed a pan-organ experimental metastasis assay where mice, under longTCP regimen, were subjected to arterial (intracardiac) injection of TCs (supplementary Fig. 5A, B). In the same way as what was observed for lung metastasis, longTCP significantly reduced metastatic outgrowth in several organs in 4T1-injected BALB/c mice (lungs, intestine, ovary, spleen, liver, kidney, brain, and the bone marrow; supplementary Fig. 5C, D). Interestingly, longTCP massively declined lung metastatic outgrowth of B16F10 (Fig. 4C, D) (45-fold), compared to shortTCP (3-fold), further emphasizing the additive effect of depleting platelets at late stages of metastasis. While lungs were equally seeded in the presence and absence of platelets (Fig. 4C, D, 15mpi and Fig. 2), longTCP completely abrogated the growth of B16F10 cells resulting in metastases-free lungs at 14dpi (Fig. 4C–F). Overall, together with the observation that platelets are passively recruited by B16F10 cells at 2hpi (Fig. 2) and that they massively populate metastatic foci at 14dpi (Fig. 3G, J), these results suggest that platelets may tune metastatic outgrowth independently of the early intravascular interaction with TCs.

To assess the role of platelets at later stages of metastatic outgrowth, we depleted platelets only once TCs had efficiently seeded the lungs and initiated metastatic growth. We decreased platelet counts from 3dpi with three successive injections of α-GPIb mAb until 10 dpi (Fig. 5A), naming this novel approach as lateTCP and thus interrogating a different contribution of platelets to metastasis (Fig. 5B). LateTCP significantly reduced B16F10 metastatic outgrowth from day 7 (Fig. 5C, D), further substantiating that platelets can control the growth of already-established and growing metastatic foci. While the number of foci in control and α-GPIb-treated mice was similar, their size was significantly reduced (Fig. 5E, F), which was related to a decline in Ki67-positive proliferating cells (Fig. 5G). As expected, we again observed that the number of intra-metastatic platelets was significantly diminished in lateTCP (Fig. 5H), suggesting that recruitment and density of platelets within metastatic foci are crucial for their outgrowth. Interestingly, the number of intra-metastatic, but not extra-metastatic, CD45-positive immune cell infiltrates were significantly reduced in TCP animals, suggesting a role of platelets in modifying the immune tumor microenvironment (TiME) towards immunosuppression (Fig. 5I), as previously shown[36]. Prompted by these observations, we performed bulk RNASeq on formalin-fixed paraffin-embedded (FFPE) samples. Transcripts related to melanoma progression and poor prognosis (*Sox6, Sox11, Tfap2a, Myo7a*) were upregulated in IgG-treated animals (Fig. 5J). This was counteracted by a positive regulation of immune cell-related genes (*Thy1, Trbc2, Cd79b, Ighg2b, Bank1, Wdfy4*) in α-GPIb-

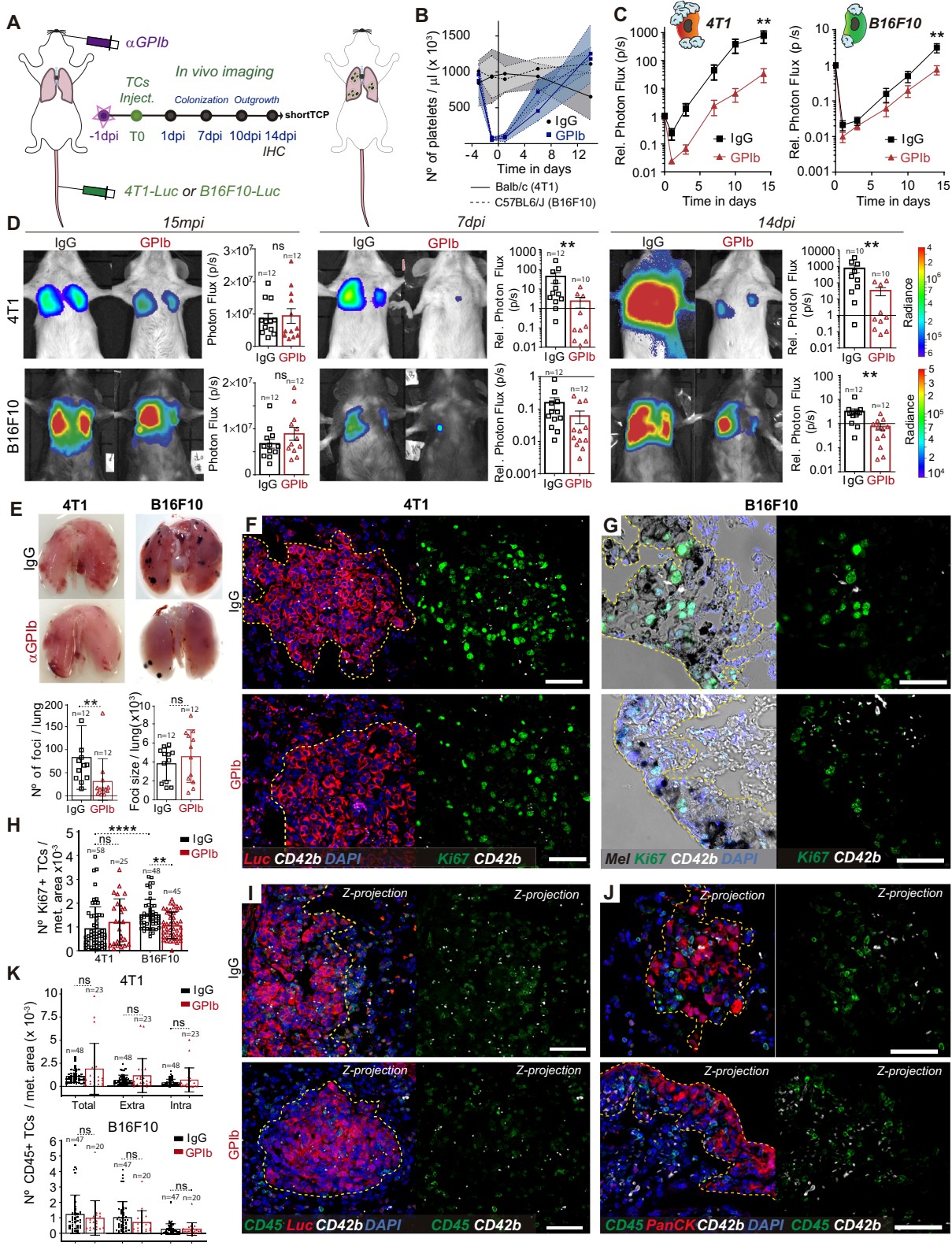

treated lungs (Fig. 5J). α-GPIb treatment also led to overexpression of genes associated with T cell co-stimulation, such as *Cd28, Cd55,* and *Nsg2*. Further analysis of upregulated genes in α-GPIb samples highlighted GO terms associated with negative regulation of IL-6 production, immune system process, and locomotion but a positive upregulation of muscle system process, myofibril assembly, and blood circulation (Fig. 5J). These results substantiate the in vivo data

(Fig. 5C–I) and show that platelets may favor transcriptional programs involved in proliferative metastatic outgrowth by controlling the immune system.

## Platelets control late steps of metastasis via GPVI

The late contribution of platelets to metastatic outgrowth opens a promising therapeutic window that could circumvent the current

**Fig. 3 | Long-term TCs metastatic potential is defined by the early TC-Platelet interplay. A** Infographics showing the experimental setting. **B** Graph showing platelet counts on IgG- and α-GPIb-treated BALB/c and C57BL6/J animals shown as mean ± SD. **C** Relative bioluminescent kinetics of 4T1-Luc and B16F10-RedLuc TCs lung seeding and outgrowth in IgG and shortTCP mice (*n* = 12 mice per groups) are shown as mean ± SEM and analyzed using two-way ANOVA with False discovery rate post-test (2 independent experiments were performed), 4T1 *q* = 0.0022, B16F10 *q* = 0.0041. **D** Representative images and timepoint (15mpi, 7dpi, and 14dpi) comparisons of total lung bioluminescence in IgG versus shortTCP animals in 4T1 (up) and B16F10 (down) TCs models. Lung bioluminescence quantifications of *n* mice (the number of mice analyzed is displayed above the histogram) are shown as mean ± SEM and analyzed using two-sided Mann–Whitney test (2 independent experiments were performed), 4T1 7dpi *p* = 0.0071, 14dpi *p* = 0.00889, B16F10 14dpi *p* = 0.0011. **E** Macroscopic analysis of mice lungs at 14dpi. Images show IgG and α-GPIb-treated lungs with 4T1 or B16F10 metastatic foci. B16F10 foci number (left) and size (right) quantifications of *n* mice (*n* = 12 per group, also displayed above the histogram) are shown as mean ± SD and analyzed using two-sided

Mann–Whitney test (2 independent experiments were performed), *p* = 0.0017. **F**, **G** Representative images from paraffin-stained IgG and shortTCP BALB/c-4T1 lungs (F) and C57BL6/J-B16F10 (**G**) at 14dpi probed against the proliferation marker Ki67 (green) and the platelet receptor CD42b (white). Scale bar: 50 µm. **H** Number of Ki67⁺ cells (number of images analyzed are displayed above the histogram) per metastatic area defined by α-pan-cytokeratin (4T1, **F**) staining or intrinsic melanin expression (B16F10, **G**) are shown as mean ± SD and analyzed using Kruskal–Wallis with False discovery rate post-test (two independent experiments were performed), *q* = 0.0001 and *q* = 0.0095. **I**, **J** Representative images from paraffin-stained IgG and shortTCP BALB/c-4T1 (**I**) and C57BL6/J-B16F10 (**J**) lungs at 14dpi probed against the hematopoietic lineage marker CD45 (green) and the platelet receptor CD42b (white). Scale bar: 50 µm. **K** Total, intra-metastatic, and extra-metastatic number of CD45⁺ cells (number of images analyzed are displayed above the histogram) per metastatic area defined by α-Luc (4T1, **I**) or α-pan-cytokeratin staining (B16F10, **J**) are shown as mean ± SD and analyzed using Mann–Whitney test (2 independent experiments were performed). Source data are provided as a Source Data file.

limitations in impairing metastatic extravasation or in detecting early metastatic foci that often occur even before cancer diagnosis[37,38]. Yet, one needs to identify the molecular targets involved and design a strategy that would avoid deleterious TCP effects while impairing the outgrowth of established metastases. To do so, we first exploited *in-house* genetic models that allow us to test the contribution of a specific platelet receptor, namely the collagen and fibrin receptor GPVI. GPVI is an immunoglobulin (Ig)−like glycoprotein that binds to many different adhesive proteins and stimulates platelet activation resulting in GPIIb/IIIa integrin activation[39]. Recent studies placed it as an attractive candidate in the promotion of platelet-dependent metastasis[40–42], mostly for the minimal impact on hemostasis. Whether it also tunes late steps of metastasis remains to be demonstrated.

To allow for syngeneicity in the GPVI⁻/⁻ mouse model and the use of two models (Fig. 6A), we engineered the mouse breast cancer AT3[41] cell line for stable luciferase expression to mirror the previously used 4T1 model (Fig. 1A) and allowed its use in a fully immunocompetent model. Interestingly, the metastatic outgrowth of AT3 cells remained unaltered in the absence of GPVI (Fig. 6B, C). Similarly, when platelets-depleting GPVI with an anti-GPVI antibody (supplementary Fig. 6B–F) that maintains a normal platelet count (supplementary Fig. 6C), lung seeding (1dpi) and metastatic outgrowth (14dpi) of 4T1 cells remained unaffected (supplementary 6C–F) suggesting that removing GPVI might be inefficient for TCs whose metastatic behavior is shaped by early intravascular platelet-TCs interactions. In contrast, the metastatic progression of B16F10 was significantly reduced in the absence of GPVI (Fig. 6B, C). While B16F10 seeding appeared unaffected (1dpi) as observed previously (Fig. 2), metastatic outgrowth was progressively impaired over time, starting from 7dpi (Fig. 6B, C), as we observed in the longTCP model (Fig. 5G, H). Macroscopic analysis of lungs revealed again that the number of foci was similar in both conditions, while their size (and proliferation index) was significantly reduced in the absence of GPVI (Fig. 6D, E). In addition, the number of intra-metastatic platelets was significantly reduced in GPVI⁻/⁻ mice (Fig. 6F). Interestingly, their activation ability, evaluated by the number of aggregates, was not affected (Fig. 6F). When probing transcriptional programs of growing metastases and surrounding stroma in both conditions, we observed, as expected, an overexpression of genes linked to pigmentation, melanin biosynthesis and cell cycle progression (DNA unwinding and mitotic spindle assembly) in WT samples, while lung metastases of GPVI⁻/⁻ presented an increase in terms associated with Ca²⁺ signaling pathway, blood circulation, monoatomic ion transportation, myofibril assembly and muscle system process (Fig. 6G). GPVI-expressing mice presented a positive z-scoring for genes related with melanoma progression and detrimental prognosis (*Mlana, Mc1r, Rab38, Cd63, Mlph, Pmel, Myo7a, Satb2*), immune checkpoint proteins (ICP) (*Cd276, Sirpa, Lgals3*) and cell cycle progression (*Cdk1, Ccnb1, Cdca2, Ccda8, Cdc20, Cdc25a, Mcm2, Mcm3, Mcm5, Top2a, Birc5*). On the other hand, GPVI⁻/⁻

mice display a positive z-score for immune cells-related genes (*Cd3g, Ebf1, Fcamr, Irf4, Ly6D, Nos1, H2-Q10*), suggesting a rewiring of the immune microenvironment at the metastatic site (Fig. 6G). Overall, these results converge with our observations in the context of lateTCP (Fig. 5J). They suggest that platelets, via GPVI, favor major colonization and metastatic outgrowth of B16F10 cells via immune suppression, as testified by the significant increase in selected immune-related genes, such as *Cd3g, Nos1, Nos3*. Also, genes related to T cells activation (*Sell*) and protection from activation-induced cell death (*Cd5*), as well as genes associated with pro-inflammatory macrophages (*Nos1, Muc5b*) and neutrophils activation (*H2-Q10, S1pr4*) were further activated in the absence of GPVI. In addition, genes related to endothelial activation (*Nos3*) and extravasation of neutrophils, macrophages, and lymphocytes to sites of inflammation (*Aoc3*) were also upregulated in GPVI⁻/⁻ samples. Altogether, this further suggests a rewiring towards a pro-inflammatory immune microenvironment at the metastatic site.

Notably, the upregulation of immune checkpoint inhibitor genes in the WT samples raises the possibility that TCs may not only proliferate faster but also manage to evade the immune system, hence leading to more efficient metastatic outgrowth. Prompted by these intriguing suggestions, we further characterize the immune microenvironment employing an ex vivo multiparametric flow cytometry approach (Fig. 6H, supplementary Fig. 7). While no difference in proportion of total CD45⁺ cells was observed (Fig. 6H), neutrophils' proportion was higher in GPVI⁻/⁻ mice (at 14dpi, Fig. 6H, and 29dpi, supplementary Fig. 6A). GPVI⁻/⁻ mice also presented a recruitment of interstitial macrophages (IM) from the bone marrow at 14dpi (Fig. 6H). Expression of *Nos1* and *Muc5b* underlined by tissue-wide RNASeq suggested that recruited IM possessed a pro-inflammatory phenotype[43,44]. Interstitial macrophage recruitment was accompanied by a decrease in lung resident alveolar macrophage proportions (Fig. 6H), whose essential role in lung metastasis has recently been highlighted[45]. B and T cell populations did not seem significantly perturbed at 14dpi while they were significantly reduced in a later timepoint (29dpi) (Fig. 6H and supplementary Fig. 6A). Altogether, these results point to an early immune response mediated by innate populations, which transitions to a preponderant effect on T and B cells at later timepoints. In addition, these data support the role of GPVI platelets' receptor in shaping metastatic outgrowth of TCs via immune suppression at the metastatic site and further emphasizes the need to consider the heterogeneity of TCs-platelet binding and molecular targets when designing platelet-targeted anti-metastatic strategies. Furthermore, they allow us to validate GPVI as an ideal target, thanks to its minimal effect on bleeding and its contribution to late steps of tumor metastasis, which makes it an appealing candidate for testing its anti-metastatic potential in a human-relevant context.

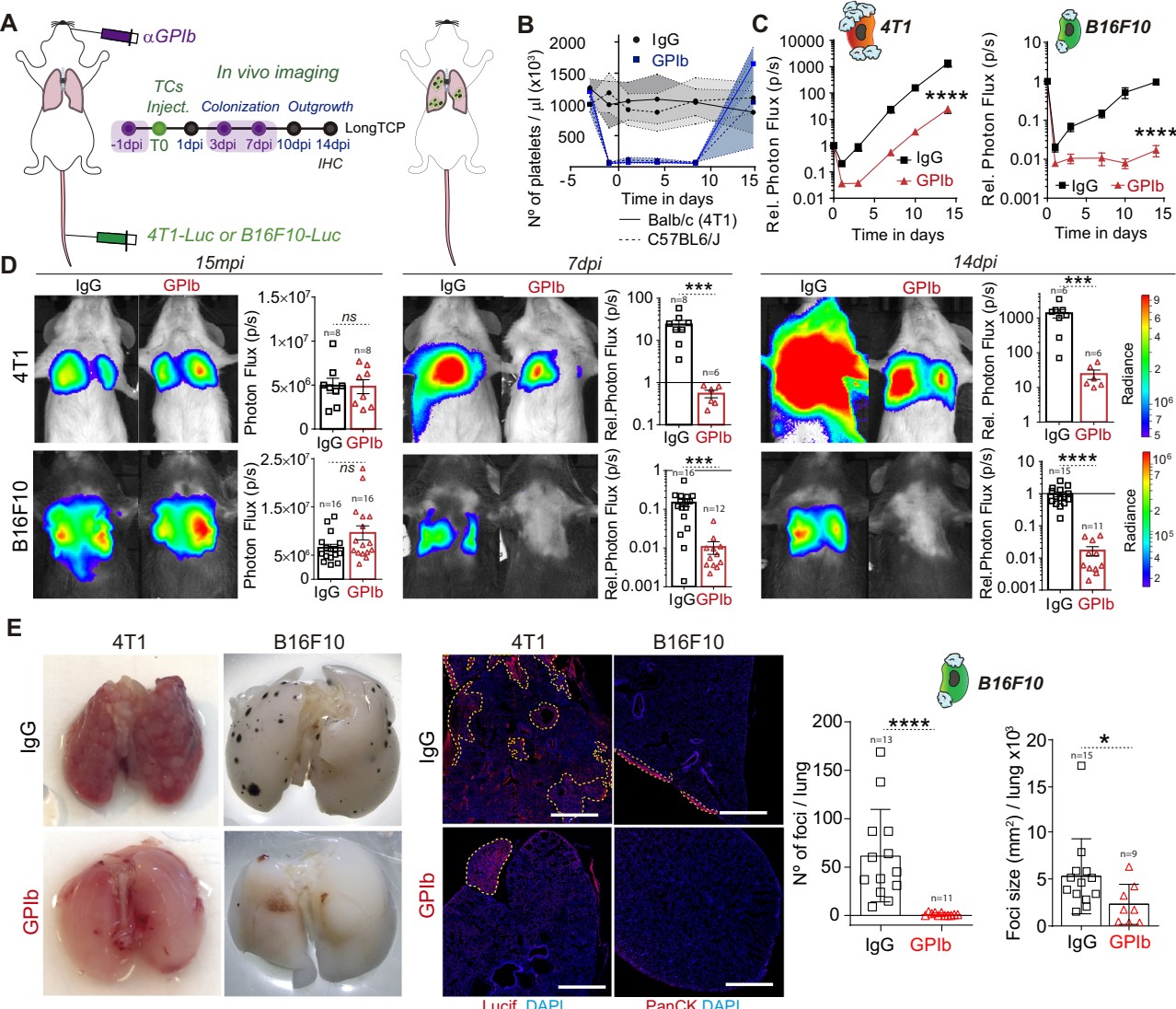

**Fig. 4 | Platelets contributes to metastatic outgrowth of already formed 4T1 and B16F10 metastatic foci. A** Infographics showing the experimental setting. **B** Graph showing platelet counts on IgG- and α-GPIb-treated BALB/c and C57BL6/J animals shown as mean ± SD. **C** Relative bioluminescence kinetics of 4T1-Luc and B16F10-RedLuc TCs lung seeding and outgrowth in IgG and long TCP mice (*n* = 8 mice per group for 4T1-Luc experiments, *n* = 16 per group for B16F10-RedLuc experiments) are shown as mean ± SEM and analyzed using two-way ANOVA with False discovery rate post-test (1 and 2 independent experiments were performed for 4T1 and B16F10 respectively), 4T1 *q* = 0.0001, B16F10 *q* = 0.0001. **D** Representative images and timepoint quantifications (15mpi, 7dpi, and 14dpi) of total lung bioluminescence in control versus long TCP animals in 4T1 (up) and B16F10 (down) TCs models. Lung bioluminescence quantifications of n mice (the number of mice analyzed is displayed above the histograms) are shown as mean ± SEM and analyzed using two-sided Mann–Whitney test (2 independent experiments were performed), 4T1 7dpi *p* = 0.0007, 14dpi *p* = 0.0007; B16F10 7dpi *p* = 0.0001, 14dpi *p* = 0.0001. **E** Macroscopic analysis of mice lungs at 14dpi (left) and low magnification confocal images of lung metastases defined by luciferase (4T1) or pan-cytokeratin (B16F10) staining at 14dpi (center). Scale bar: 500 µm. B16F10 foci number and size (number of mice analyzed per group are displayed above the histogram) are shown as mean ± SD (right panel) and analyzed using two-sided Mann–Whitney test (2 independent experiments were performed), foci number *p* = 0.0001, foci size *p* = 0.029. Source data are provided as a Source Data file.

## Targeting the human GPVI receptor with a humanized antibody impairs metastasis

While this study and others[40] confirm that GPVI is a promising target for impairing metastasis, its relevance to human pathology is missing. Therefore, better preclinical models and humanized therapeutic tools, with ideally no TCP effects, are needed. We first interrogated whether platelets would also populate human metastatic foci. With that aim, we analyzed human lung metastasis biopsies from metastatic melanoma patients and found detectable intra-metastatic platelets (Fig. 7A), providing the prime evidence that platelets can populate human metastatic foci and further suggesting that they may control metastasis at the late stages of their progression. We then exploited our previously established genetically modified mouse strain human GPVI (hGPVI)[42], which bears a knock-in copy version of human GPVI without presenting any viability or hematological defects, making it the best candidate to test α-GPVI compounds. To counteract GPVI function, we exploited glenzocimab (ACT017, Acticor Biotech)[46], a humanized antibody fragment (Fab) that has recently completed a phase II trial in stroke patients and is currently examined in a phase II/III trial (ACTISAVE, ClinicalTrials.gov Identifier: NCT05070260). This compound binds to hGPVI to block ligand binding, without causing thrombocytopenia or GPVI membrane depletion, and it is not associated with bleeding events[47].

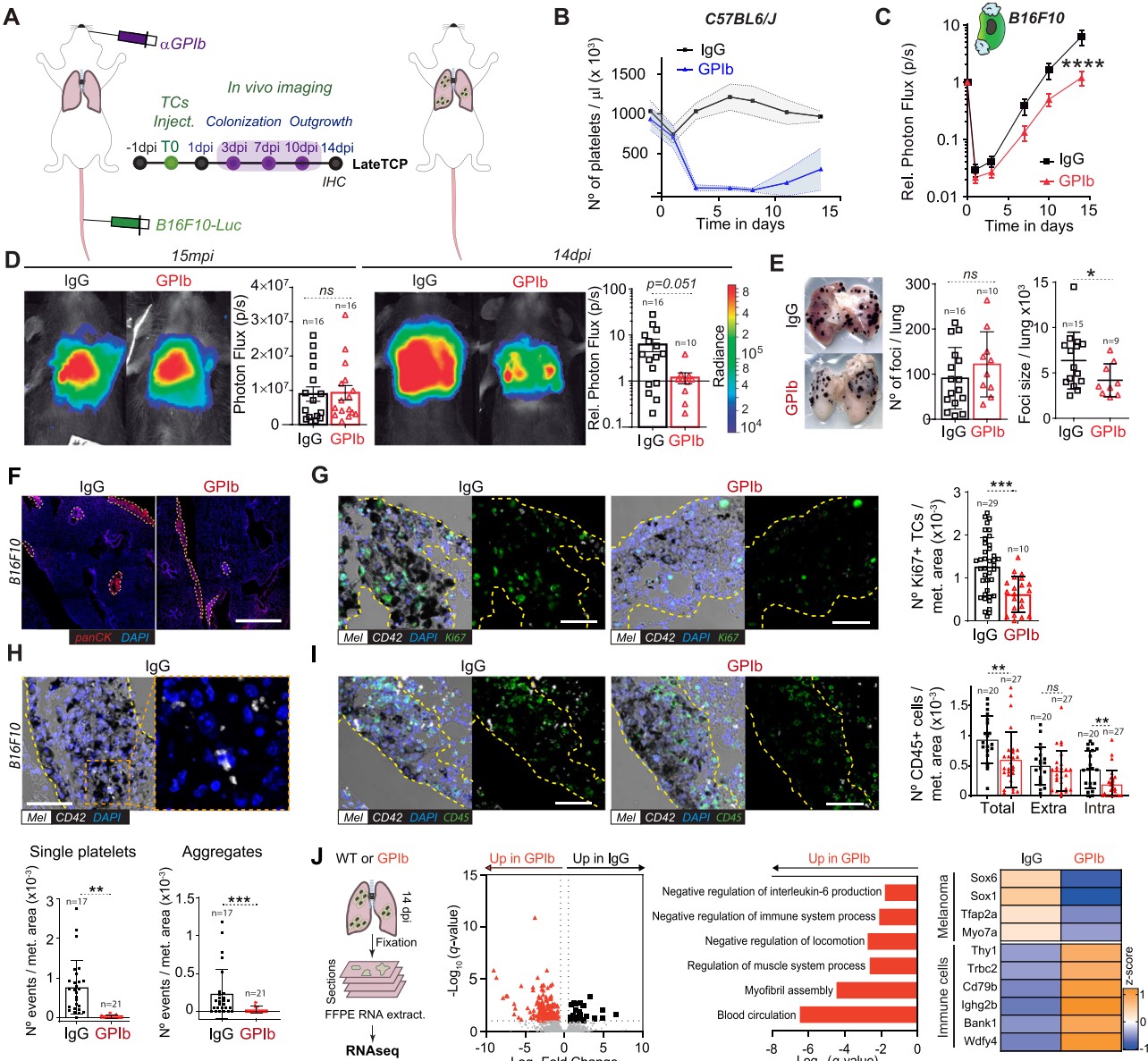

**Fig. 5 | Platelets significantly influence the late outgrowth of B16F10 metastatic foci but not 4T1 ones. A** Infographics of the experimental scheme. **B** Graph of platelet counts of IgG- and α-GPIb-treated animals are shown as mean ± SD. **C** Relative bioluminescence kinetics of B16F10-RedLuc TCs lung seeding and outgrowth in IgG and late TCP mice (n = 16 per group) are shown as mean ± SEM and analyzed using two-way ANOVA with False discovery rate post-test (2 independent experiments were performed), p = 0.0001. **D** Representative images and timepoint comparisons (15mpi and 14dpi) of total lung bioluminescence in control versus late TCP animals in B16F10 TCs models. Lung bioluminescence quantifications of n mice (the number of mice analyzed is displayed above the histogram) are shown as mean ± SD and analyzed using two-sided Mann–Whitney test (2 independent experiments were performed), 14dpi p = 0.0513. **E** Macroscopic analysis of mice lungs at 14dpi. Images show IgG- and α-GPIb-treated lungs with B16F10 metastatic foci (number of mice analyzed per group are displayed above the histogram). B16F10 foci number and mean foci size are presented as mean ± SD and analyzed using two-sided Mann–Whitney test (2 independent experiment were performed), foci size p = 0.0401. **F** Low magnification confocal images of lung metastases defined by pan-cytokeratin staining at day 14 post injection. Scale bar: 500 μm. **G–I** Immunohistochemical analysis of IgG and late TCP mice lungs at 14dpi. **G** Representative images of lung sections from IgG and late TCP B16F10 probed against the proliferation marker Ki67 (green), the platelet receptor CD42b (white) and intrinsic melanin expression (black). Scale bar: 50 μm. Total number of images

analyzed are displayed above the histogram. Number of Ki67+ cells per metastatic area are shown as mean ± SD and analyzed using two-sided Mann–Whitney test (2 independent experiments were performed), p = 0.0097. **H** Representative images of lung sections from IgG and late TCP B16F10 probed against the platelet receptor CD42b (white). Scale bar: 50 μm. Total number of images analyzed are displayed above the histograms. Single platelet (≤3) and platelet aggregates (>3) per metastatic area are shown as mean ± SD and analyzed using two-sided Mann–Whitney test (2 independent experiments were performed), single p = 0.0065, aggregates p = 0.0002. **I** Representative images of lung sections from IgG and late TCP B16F10 probed against the hematopoietic lineage marker CD45 (green), the platelet receptor CD42b (white), and intrinsic melanin expression (black). Scale bar: 50 μm. Total, intra-metastatic, and extra-metastatic number of CD45+ cells per metastatic area defined by melanin are shown as mean ± SD and analyzed using Mann–Whitney analysis (2 independent experiments were performed), total p = 0.0016, intra p = 0.0058. **J** FFPE-RNASeq analysis. From the left: representative infographics of the experimental scheme; volcano plot of genes differentially regulated in IgG and α-GPIb-treated mice; GO terms assignment of α-GPIb-derived samples upregulated genes; z scoring comparing IgG and α-GPIb samples for selected genes. Data are representative of three mice per group from two independent experiments. Source data are provided as a Source Data file.

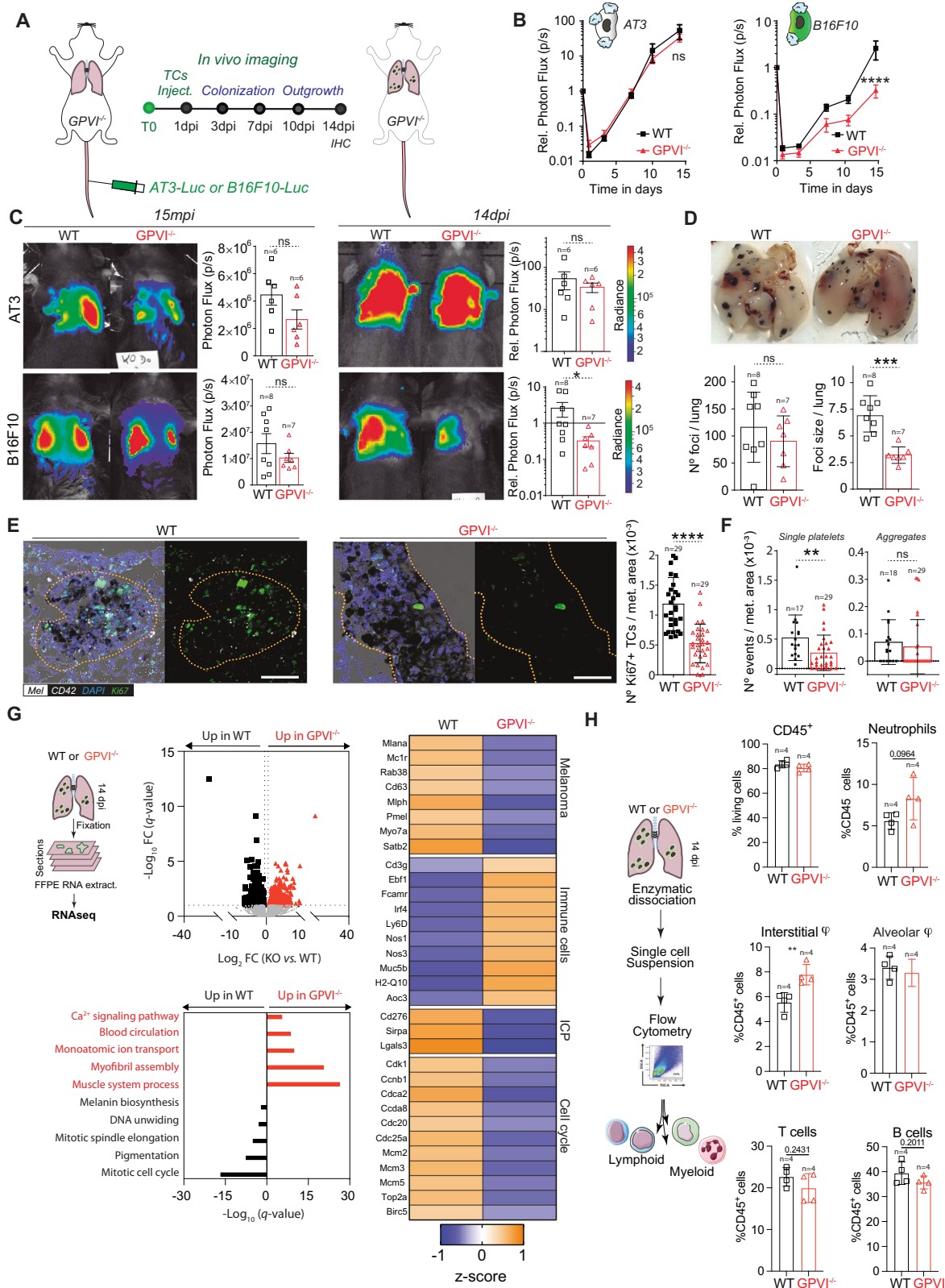

Building on the exciting observation that lateTCP could reduce the growth of established metastases (Fig. 5), we subjected hGPVI mice to an experimental metastasis protocol targeting the growth of already formed metastases (mimicking lateTCP, Fig. 5G). The half-life of glenzocimab in mice is rather short when compared to humans[47]. We thus administered glenzocimab progressively using osmotic pumps that were implanted subcutaneously (Fig. 7B). We confirmed that

glenzocimab did not affect platelet counts (Fig. 7C) and was found to be successfully bound to mouse platelets (Fig. 7D) fourteen days after pump implantation. Longitudinal imaging of B16F10 lung metastasis revealed that the metastatic burden of glenzocimab-treated hGPVI mice was significantly reduced starting from 4 days after the implantation of the pump (Fig. 7E, F), reaching a four-fold decrease at 14dpi (Fig. 7E). Macroscopic inspection of the lungs manifested that

**Fig. 6 | GPVI receptor-mediated platelets' metastatic outgrowth effect.**
**A** Infographics showing the experimental setting. **B** Relative bioluminescence
kinetics of AT3-RedFluc-L2 and B16F10-RedLuc TCs lung seeding and outgrowth in
WT and GPVI$^{-/-}$ mice ($n = 6$ mice per group for AT3, $n = 7$ or 8 respectively for WT or
GPVI$^{-/-}$ group for B16F10) are shown as mean ± SEM and analyzed using two-way
ANOVA with False discovery rate post-test (1 experiment was performed), B16F10
$q = 0.0001$. **C** Representative images and timepoint comparisons (15mpi, left; 14dpi,
right) of total lung bioluminescence in WT versus GPVI$^{-/-}$ depleted animals in AT3
(up) and B16F10 (down) TCs models. Lung bioluminescence quantifications of $n$
mice (the number of mice analyzed is displayed above the histogram) are shown as
mean ± SD and analyzed using two-sided Mann–Whitney test (1 experiment was
performed), B16F10 14dpi $p = 0.0289$. **D** Macroscopic analysis of mice lungs injec-
ted with B16F10 cells at 14dpi. Pictures of WT (left) and GPVI$^{-/-}$ (right) lungs are
show. B16F10 foci number and size are shown as mean ± SD and analyzed using two-
sided Mann–Whitney test (1 experiment was performed, number of mice analyzed
per group are displayed above the histogram), foci size $p = 0.003$. **E** Representative
zoomed images of the immunohistochemical analysis of lung tissue sections of WT
and GPVI$^{-/-}$ mice probed against the proliferation marker Ki67 (green), the platelet

receptor CD42b (white), and intrinsic melanin expression (black). Scale bar: 50 µm.
Total number of lung sections analyzed is displayed above the histogram. Number
of Ki67$^+$ cells per metastatic area are shown as mean ± SD and analyzed using two-
sided Mann–Whitney test (1 experiment was performed), $p = 0.0001$. **F** Number of
intra-metastatic platelets (from **E**, total number of lung sections analyzed are dis-
played above the histogram) per metastatic area in WT and GPVI$^{-/-}$ animals are
shown as mean ± SD and analyzed using two-sided Mann–Whitney test (1 experi-
ment was performed), $p = 0.0045$. **G** FFPE RNASeq analysis. From the left: repre-
sentative infographics of the experimental scheme; volcano plot of genes
differentially regulated in WT and GPVI$^{-/-}$; GO terms assignment of WT and GPVI$^{-/-}$
upregulated genes; $z$-scoring comparing WT and GPVI$^{-/-}$ samples for selected
genes. Data are representative of 3 mice per group from 2 independent experi-
ments. **H** Ex vivo lungs immunophenotyping. Left: Infographics describing the
experimental scheme. Right: percentage of relative immune cells populations as
assessed by flow cytometry analysis 14dpi. Data are represented as mean ± SD and
analyzed using two-sided Student test ($n = 4$ mice per group, 1 experiment was
performed, interstitial macrophage $p = 0.0078$). Source data are provided as a
Source Data file.

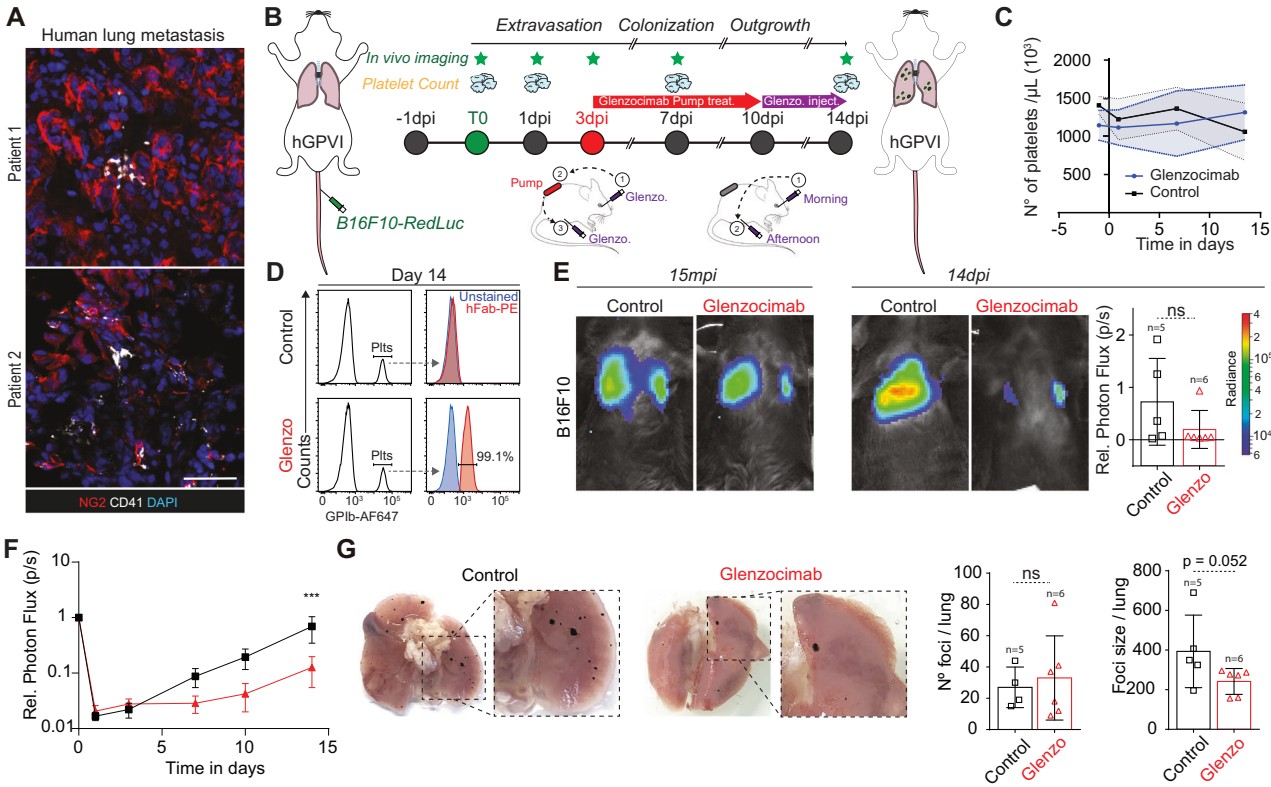

**Fig. 7 | Therapeutic targeting of the human GPVI receptor impairs metastatic
outgrowth of established foci.** **A** Representative confocal images of human
biopsies from melanoma metastases (patient1 and patient2) probed against the
melanoma marker NG2 (red) and the platelet marker CD41 (white). Scale bar:
50 µm. **B** Infographics showing the experimental setting. Glenzocimab administra-
tion is performed via subcutaneous osmotic pumps from 3dpi to 10 dpi (red
arrow) then continued by morning retro-orbital (1) and afternoon intra-peritoneal
(2) injection until 14dpi. **C** Graph showing platelet counts in control and
Glenzocimab-treated animals ($n = 5$ mice in the control group, $n = 6$ in the Glen-
zocimab group) and shown as mean ± SD. **D** Flow cytometry analysis of mouse
platelets labeled ex vivo. Left: electronic gating strategy on platelets labeled with α-
GPIb-647. Right: ex vivo platelets staining with anti-human (Fab)–PE-labeled anti-
body at 14dpi. **E** Representative images and timepoint (15mpi and 14dpi) compar-
isons of total lung bioluminescence in control versus Glenzocimab-treated animals.

Day 14 lung bioluminescence quantifications of $n$ mice (the number of mice ana-
lyzed are displayed above the histogram) are shown as mean ± SD and analyzed
using two-sided Mann–Whitney test (2 independent experiments were performed).
**F** Relative bioluminescence kinetics of B16F10-RedLuc TCs lung seeding and out-
growth in isotype or glenzocimab-treated mice ($n = 5$ mice in the control group,
$n = 6$ in the Glenzocimab group) are shown as mean ± SEM and analyzed using two-
way ANOVA with False discovery rate post-test (2 independent experiments were
performed), $q = 0.0007$. **G** Macroscopic analysis of lungs from mice injected with
B16F10 cells at 14dpi. Pictures of control (left) and glenzocimab-treated (right)
lungs are shown. B16F10 foci number and size are shown as mean ± SD and analyzed
using two-sided Mann–Whitney test (2 independent experiments were performed,
number of mice analyzed per group is displayed above the histogram). Source data
are provided as a Source Data file.

glenzocimab had, as expected since the treatment started after seeding, no effect on the number of metastatic foci, but reduced their size (Fig. 7G). Altogether, these results identify, distinctly, an α-GPVI compound able to target human GPVI and impair metastatic progression without side effects (thrombocytopenia, bleeding). Considering the very positive safety profile of glenzocimab and its efficiency in repressing growing metastasis, we provide evidence for the repositioning of this therapeutic agent with previously unmet clinical potential.

## Discussion

This manuscript reports a pioneering longitudinal analysis of the pro-metastatic role of platelets in lung metastasis by subjecting in parallel two complementary TCs models to experimental metastasis assays in the zebrafish embryo and mice. The two TCs were tested in six mouse models of time-controlled thrombocytopenia and by the genetic and pharmacological blocking of the platelet receptor GPVI. We report that different TCs interact with platelets differently, and confirmed that such ability can, in part, shape intravascular arrest and metastatic fitness of TCs. Importantly, we provide an original study demonstrating that platelets contribute to late steps of metastasis and document the therapeutic efficacy of targeting GPVI in animals carrying growing metastases, an experimental approach never tested before. Altogether, our data identify GPVI as the main molecular target whose inhibition can impair metastasis without inducing collateral hemostatic perturbations.

The established mechanisms proposed for the pro-metastatic role of platelets (including platelet binding, activation, and aggregation by TCs) have been traditionally insufficient in providing momentum for the development and establishment of anti-platelet drugs in routine oncologic care. Options for therapeutic intervention are limited by the narrow and early window within the metastatic process where platelets interact with CTCs. Surprisingly, one assumes that all cancers interact equivalently with platelets, leading to controversial results in different cancer models[23,24] and unclear outcomes when using anti-platelet drugs in oncologic patients[25,48]. Indeed, while CTCs are widely considered to be covered by platelets when entering the blood[18], whether this applies to all cancer types remains to be demonstrated. Here, we show that TCs do not all bind platelets equally (Fig. 1) and, importantly, that TCIPA seems to occur only at high concentrations of TCs in vitro, which is an unlikely scenario in vivo[26,49,50]. In this regard, thrombocytosis has been documented as a poor prognosis factor, reflecting an aggressive tumor phenotype in cervical, colon, and non-small cell lung cancer patients[51,52]. On the contrary, in pancreatic cancer, thrombocytopenia was associated with a worse prognosis, thus indicating the varied influence of platelets depending on the type of cancer[53,54]. All these issues have left anti-platelet drugs out of the standard cancer treatments and/or prophylaxis. In order to provide further insights into platelets' pro-metastatic role also in the late stages of the disease, we developed in this study a novel experimental approach merging the single cell analysis of in vivo arrested TCs combined with in vivo bioluminescence longitudinal analysis of micro- and macro-metastases in mouse models of time-controlled thrombocytopenia. Thanks to this, we were able to identify GPVI platelet receptor as a mediator of this platelets' function, and we probed the feasibility of its targeting with a therapeutic antibody.

We first explored the pro-metastatic effect of platelets at early stages and found that platelets favor the survival of intravascularly arrested TCs by direct binding to the surface of viable cells or by passive recruitment around non-viable cells. This passive recruitment may occur by vessel occlusion or release of platelet-activating cytoplasmic content such as tissue factor (TF)[20,55]. While direct binding significantly supports long-term metastatic fitness (Fig. 3), passive recruitment of platelets by flow clogging further supports metastasis by impacting late outgrowth behavior (14dpi). Previous work has

suggested a key role for thrombin and clot formation as the starting point of brain metastasis[20], however, not all metastatic cell lines express TF. Considering that metastasis is a very inefficient process, and that intravascular cell death is a well-known cause of TF release and thrombin production, the co-option of viable with non-viable cells may help the viable cell to successfully metastasize, as previously observed[56]. Such ability of platelets to be passively recruited to metastatic cells may be capital for designing anti-platelet drugs for treating different cancer types.

While previous reports have found evidence of platelets inside primary tumors[11,32,33], no evidence proves that this could apply to metastasis. We have now documented that metastatic sites are populated by platelets suggesting that they might be able to extravasate with or independently from TCs, as reported previously[57]. Platelets have been shown to migrate within tissues[58], suggesting that they could actively populate growing metastases, but whether they require extravasation to seed metastases remains to be demonstrated, as they might just interact and imprint TCs in circulation. For example, while TCs interact with platelets, fibrin, or fibrinogen from 0 to 6 h post injection, platelets are not found at intravascular arresting sites from 6 to 24hpi[54], meaning that early intravascular interactions are frugal. In this sense, the GPVI receptor may tune metastasis by passive and intermittent interaction with TCs at the arrest sites[59], similar to what is shown by our genetic and pharmacological loss-of-function models at late steps of the cascade. Furthermore, the observation that platelets' number but not their aggregation capacity is affected in GPVI-KO animals could hint towards migration deficiencies in this model and open new areas of investigation.

Platelets are known to release significant amounts of bioactive material within releasates[60] that shape, for example, their invasion abilities. In addition, they also secrete microparticles and extracellular vesicles, with angiogenic[61], immunosuppressive[10], and pro-survival abilities[62] capacity, as hinted in this study (increased proliferative index in the presence of platelets in our short and lateTCP models). These observations were further validated by RNASeq on FFPE lung sample where genes associated with melanoma progression and proliferation were overexpressed in IgG-treated animals (Fig. 5J) or repressed in our in-house GPVI$^{-/-}$ model (Fig. 6G).

Besides the early shielding, and protecting CTCs from immune cells during circulation, platelets profoundly impact the TiME. Indeed, we demonstrated here that platelets massively infiltrate lung metastatic lesions and are activated by B16F10 TCs (Fig. 3G, F and supplementary Fig. 4B, C). Upon activation, platelets release large quantities of immunomodulatory molecules (TGFβ, PGE2, maresin-1, etc.) whose roles in T cells polarization modulation, anti-inflammatory macrophages induction, myeloid-derived suppressor cells (MDSCs), N2 neutrophils and/or tolerogenic dendritic cells recruitment has been previously demonstrated[63,64]. Moreover, platelets express class I MHC conferring a "pseudonormal" phenotype to cancer cells and subverting NK cells anti-tumoral functions[65]. They are also able to convert ATP in adenosine, a potent immunosuppressive molecule, through membrane expression of the ectonucleotidases CD39 and CD73[66]. Platelet's role in pro-tumoral, immunosuppressive TiME promotion was also confirmed in our models. Indeed, RNASeq of α-GPIb-treated lungs highlighted an increase in immune cell-related transcripts (Fig. 5J). Furthermore, our data suggest that platelets-driven immune suppression is in part mediated by GPVI as demonstrated by RNASeq and ex vivo lung immunophenotyping of our in-house GPVI$^{-/-}$ mouse model. In this study, we underlined that GPVI deletion was associated with higher immune cell transcript expression (Fig. 6G) and recruitment of bone marrow-derived interstitial macrophages (IM) presenting a pro-inflammatory gene signature (Fig. 6G, H). On the other hand, mice expressing GPVI showed increased expression of immune checkpoint inhibitor genes (Cd279, Sirpa). Overexpression of galectin-3 (Lgals3), a known GPVI ligand[59], in metastatic lungs further supports

the implication of GPVI in the TiME remodeling. In addition, recent meta-analyses underpinned the association of high platelets to lymphocytes ratio (PLR) with dismal response to chemotherapy and poor overall survival in clinical settings[67,68]. Based on these pieces of evidence, we might speculate that platelets influence metastatic outgrowth by (i) promoting cancer cell proliferation and (ii) rewiring the TiME towards an immunosuppressive one. Nonetheless, additional investigations are needed to deepen our observations, notably on the cellular functions of the metastatic TiME, their associated transcriptional program, and local cytokines secretion. Given these exciting indications, reduction of platelets' count seems an appealing approach to hamper metastatic burden. Nevertheless, the augmented risk of bleeding due to their depletion makes this approach not feasible, instead boosting the targeting of platelets' receptors to circumvent this issue. In this line, our study shows that depletion or targeting of platelets' GPVI successfully impairs the proliferation rate within B16F10 metastatic foci with no collateral damage. These data reinforce the potential of targeting GPVI in cancer as shown in previous studies where it already leads to decreased intratumor hemorrhage and improved intra-tumoral administration of chemotherapeutic agents[17]. Overall, this points toward GPVI as a promising candidate for the development of new anti-metastatic drugs able to address cancer once the metastatic spread is already initiated, which accounts for most of the cases in clinical practice. Such achievement is of utmost importance considering that we also inform here on how a pharmacologic blockade of GPVI with glenzocimab (Acticor Biotech), an anti-GPVI F(ab) fragment evaluated in phase I clinical trials to prevent ischemic stroke, efficiently reduces experimental metastasis. As most of the compounds available in the market are associated with increased risk of hemorrhage in clinical settings or limited efficacy against metastasis, with the exception of aspirin[69], this finding results particularly promising as glenzocimab presents an anti-thrombotic effect without inducing any bleeding in healthy volunteers and in patients[70]. Altogether, our work demonstrates that targeting platelets in a progressively metastatic animal model can impair their growth, and it further identifies GPVI as the key molecular target whose inhibition can hamper metastasis without inducing collateral hemostatic perturbations.

## Methods

### Cell lines

The mouse breast cancer cell lines 4T1, E0771, AT3, and D2A1, and the mouse melanoma cell line B16F10, were cultured under standard conditions (37 °C, 5% CO$_2$) using RPMI-1640 or DMEM, respectively, supplemented with 10% FBS and 1% penicillin-streptomycin solution. The human breast cancers MDA-MB-231, MCF7, and melanoma cell lines A375, we cultured under the same conditions (DMEM). Their viabilities in vitro were assayed before aggregation and SEM assays and in vivo experimental metastasis experiments by an ADAM-MC Automated cell counter (ThermoFisher).

### Cell line engineering

For in vivo imaging on zebrafish embryos, 4T1 and B16F10 cells were engineered to express a Lifeactin-tdTomato fusion protein. Briefly, the Life-tdTomato DNA fragment from the Addgene plasmid tdTomato-Lifeact-7 (number 54528) was inserted in the pLSFFV-Ires-Puromycin or pLenti-CMV-mPGK-Puromycin lentiviral vectors to generate the pLSFFV-LifeActin-tdTomato-Ires-Puro and the pCMV-LifeActin-tdTomato-mPGK-Puro lentivirus lentivirus vectors used to transduce 4T1 and B16F10, respectively. Lentivirus particle production and transduction procedures have been previously described[30]. For confocal imaging of single arrested TCs in mice lungs, luciferase-expressing 4T1 or B16F10 cells were labeled in vitro with CellTrace Yellow (Thermo Fischer) according to manufacturer's instructions.

For in vivo imaging in mice, regular firefly luciferase (*Photinus pyralis*) (4T1-Luc, a kind gift from Corinne Laplace-Builhé, Institut Gustave Roussy Paris) or red-shifted firefly luciferase (Bioware® Brite B16F10-RedFLuc, PerkinElmer) was used. A luciferase-expressing breast cancer AT3 cell line[41] was engineered in-house with RediFect RedFluc Lentiviral Particles (PerkinElmer). To avoid immune rejection in immunocompetent C57BL6/J recipients, an in vivo-derived AT3 cell line was established from murine lung metastases. Briefly, mouse lungs containing AT3-RedFLuc metastasis were collected on day 18 post injection (dpi) from tail vein injected animals. Tissue was chopped into 1 mm$^3$ pieces with a scalpel, resuspended in 4000 μL of dissociation buffer (Miltenyi Biotech Neural Tissue Dissociation Kit (P) 130-092-628) and mild enzymatically digested with 2.5 mg/ml collagenase solution (Sigma-Aldrich C9891) and 400 μg/mL DNaseI under gentle mixing at 37 °C for 10 min. Enzyme P (50 μL) and enzyme A (60 μL) were sequentially added and incubated 5 minutes each. The cell suspension was filtered through a 40 μm cell strainer (Corning, 431750) before centrifugation for 5 min, 400 × $g$ at room temperature (RT). The cell-containing pellet was plated in DMEM medium + 10% FCS + 1% PS + 1% Gentamicin + 5 μg/ml Fungizone + 1 μg/ml puromycin in a T75 flask. After a few days of puromycin selection, the cell line was considered established (AT3-RedFluc-LM1). To ensure the absence of immune rejection on immunocompetent recipients a second round of in vivo derivation was performed, thus generating the AT3-RedFluc-LM2 cell line, further used in our experiments.

### Platelets' isolation

Human citrated platelet-rich plasma (cPRP) was obtained from supernatant derived from citrate buffer (3.8%) collected blood centrifuged 250 × $g$ for 16 min at room temperature. Washed human platelets suspensions were obtained under informed consent from healthy donors not undergoing any platelets-affecting drug treatment for the previous 10 days. Blood was collected in the presence of 1/7 volume of citric acid-citrate-dextrose (ACD) pH 6.5, and washed platelets were prepared as described previously[71]. The quality of preparations was routinely screened in a light aggregometry assay with ADP 5 μM as aggregation inductor in the presence of fibrinogen (for washed platelets) or in its absence (for cPRP).

Murine citrated platelet-rich plasma (cPRP) blood was collected by aortic puncture in citrate buffer 0.315% following centrifugation of the whole blood in microtubes during 1 min at 1900 × $g$ at RT. Murine washed platelet suspensions were obtained from aortic puncture in anesthetized mice (intra-peritoneal, i.p., ketamine 100 mg/kg, xylazine 20 mg/kg). Blood (700–800 μL) was collected in an ACD-preloaded syringe by puncture at the level of the separation of the aortic artery into the two iliac arteries. Mice were sacrificed by cervical dislocation at the end of the procedure.

### Clusters formation and anoikis resistance assay

$2 \times 10^6$ 4T1 or B16F10 cells in suspension were co-incubated with $2 \times 10^7$ murine cPRP for 2 hours at 37 °C 5%CO$_2$ on a tilting tray. Cells were then washed with serum-free medium, and clusters were imaged on an inverted microscope. Macroscopic RGB images with identical zooming were loaded on ImageJ, and clusters' area was measured after polygon selection definition.

Response to anoikis was determined in clustered cells plated at 500,000 cells/mL in sixwell ultra-low adherence plate in complete culture medium for 24 h or 48 h later. At the end of the incubation, cells were harvested and stained with Zombie NIR (Biolegend, Cat.423105) (1:1000) for 15 minutes at RT in the dark, prior to wash and signals' acquisition on an Attune NxT (Invitrogen) flow cytometer (20,000 events/conditions). Data were analyzed using FlowJo™ v10 Software (TreeStar).

## Zebrafish handling and intravascular injection of TCs

*Tg(fli1a:eGFP)* Zebrafish (*Danio rerio*) embryos from a Tubingen background were kindly provided by the group of F. Peri from EMBL (Heidelberg, Germany) and further grown and bred in our in-house zebrafish facility. Embryos were maintained in Danieau 0.3× medium (17.4 mM NaCl, 0.2 mM KCl, 0.1 mM MgSO4, 0.2 mM Ca(NO3)2) buffered with HEPES 0.15 mM (pH = 7.6), supplemented with 200 μM of 1-Phenyl-2-thiourea (Sigma-Aldrich) to inhibit the melanogenesis, as in[72]. Forty-eight-hour post-fertilization (48hpf) embryos were mounted in a 0.8% low melting point agarose pad under 650 μM tricaine (ethyl-3-aminobenzoatemethanesulfonate) anesthesia. LifeAct-TdTomato TCs were injected with a Nanoject microinjector 2 (Drummond) and micro-forged glass capillaries (25–30 μm inner diameter) filled with mineral oil (Sigma). For the intravascular injection of TCs, $100 \times 10^6$ TCs/mL 4T1 or B16F10 were washed (EDTA 0.48 mM, Versene, Gibco) and gently detached (Trypsin solution 0.125% diluted in Versene 1×, Gibco), then resuspended in serum-free RPMI-1640 or DMEM medium and maintained in ice until injection. Prior injection, 50 μL TCs:50 μL human cPRP (350-450,000 platelets/μL (1/4 ratio approx.)) were mixed and incubated at 37 °C for 2-3 minutes. Human citrated platelet poor plasma (cPPP) was used as a negative control. Finally, 23 nL of the TCs-platelets mix was injected in the duct of Cuvier under the M205 FA stereomicroscope (Leica), as in ref. 73.

## Mice

8–16 weeks male WT C57BL6/J immunocompetent mice were used for the B16F10 melanoma cell model, while female WT immunocompetent BALB/c mice were used for the 4T1 breast cancer model. All mice were purchased from Charles River Labs. C57BL6/J mice deficient for the platelet receptor GPVI (GPVI$^{-/-}$), and the humanized GPVI model (hGPVI) were generated as previously described[42]; males and females were used for both models. Animals were housed in pathogen-free conditions with food and water *ad libitum* and appropriate enrichment (sterile pulp paper and coarsely litter). Mice were monitored daily, and terminally sick animals were euthanized under an approved protocol. All animal procedures were performed in accordance with institutional guidelines and approval, under the APAFIS authorization 14741-2018041816337540.

## Experimental pulmonary metastasis assay and bioluminescent imaging

Subconfluent 4T1-Luc or B16F10-RedFluc were washed with EDTA 0.48 mM Versene, Gibco), gently detached using a 0.25% trypsin-0.02% EDTA solution (Gibco), washed in media containing 10% FBS, resuspended at $1.5 \times 10^6$ TCs/mL in serum-free media, filtered through a 40 μm mesh (Falcon) and kept on ice until injection. Viability (>85% prior injection) was determined by trypan blue exclusion. TCs (100 μL) were injected through the lateral tail vein (intra-venously, i.v.) with a 25-gauge needle. In vivo imaging was performed shortly after to establish initial lung seeding (15mpi) by i.p. injection of D-luciferin solution (150 mg/kg) and subsequent (5 minutes later) signal acquisition of mice (isofluorane-anesthetized - Isoflo, Zeotis) ventral view with an IVIS Lumina III (PerkinElmer) imaging system. The rate of total light emission of the lung metastatic area was analyzed using the Living Image software (PerkinElmer) and expressed as numbers of photons emitted per second (p/s). Subsequent imaging points at 0.5, 1, 2, 4, and 24 h (initial homing) and 1, 3, 7, 10, and 14 days (outgrowth) after TCs injection were monitored. Dynamics of homing and extravasation appeared similar (around 24hpi) when the outgrowth of the two cell lines was assessed. For later timepoints (1, 3, 7, 10, 14 days), the relative total light emission of the lung metastatic area compared to day 0 was calculated and represented.

## Intracardiac cells injection

$5 \times 10^6$ 4T1-Luc/mL suspension in PBS was prepared as aforementioned. 500,000 cells were injected in the left ventricle with a 25-gauge needle in anesthetized mice placed in ventral position. In vivo imaging and analysis were performed as before. At the time of the sacrifice, ex vivo signal was measured from harvested organs (lungs, intestine, ovary, spleen, liver, kidney, brain, bone marrow), and the rate of total light emission was analyzed and expressed as numbers of photons emitted per second (p/s).

## Platelet and GPVI receptor depletion and pharmacological blockage

Severe thrombocytopenia was induced by i.v. injection of 2 mg/kg of in-house-produced α-GPIb antibody (RAM.6)[74] or IgG isotype control per mouse. Injections were set according to the experimental scheme: −1dpi TCs for shortTCP; -1/3/7dpi TCs for LongTCP; 3/7/10 dpi TCs for LateTCP. Murine GPVI platelets' receptor was antagonized by 50 μg of JAQ1 mAb (1 mg/mL, Emfret) or rat IgG isotype control (Emfret) subcutaneously (s.c.) injected per mouse. Anti-human GPVI antibody Glenzocimab (ACT017, Acticor Biotech[46]) was delivered through an Azlet osmotic pump (model 2001) implanted s.c. on mouse's dorsal side (APAFIS authorization 37433-2022052016445806). Pump delivery was maintained for 7 days at 1 μl/hour with a concentration of 20 mg/ml. After, the treatment was continued for an additional 3 days with 10 mg/mL/serum physiological retro-orbitally injected per mouse in the morning and 120 μL/mouse i.p. injected in the afternoon.

## Platelet counts

Murine whole blood was collected into EDTA (6 mM) after severing the mouse tail. The platelet count and size were determined in a Scil Vet abc automatic cell counter (Scil Animal Care Company, Holtzheim, France). The platelet count on human platelet suspensions (cPRP and/ or washed platelets) was assessed in an automatic platelet counter (XN-1000, Sysmex).

## Ex vivo flow cytometry

Single-cell suspension was obtained from freshly harvested left lung lobes. Tissue was cut into approximately 2mm$^3$ pieces using a scalpel, then enzymatically digested with Tumor Dissociation kit, mouse (Miltenyi, 130-096-730) at 37 °C for 42 minutes using the gentleMACS Octo Dissociator (Miltenyi, 130-096-427) according to manufacturer's protocol. Cell suspension was filtered through a 50 μm mesh and red blood cells were lysed in ACK Buffer [150 mM NH$_4$Cl, 10 mM KHCO$_3$, 0.1 mM Na$_2$EDTA]. Non-viable cells were stained with Fixable Viability Dye-eFluor™780 (eBioscience™, 65-0865-14) for 15 minutes at room temperature in the dark. Aspecific antibodies' binding was minimized with anti-CD32/CD16 blocking antibody (TruStain FcX™ 1:50, Biolegend, Cat.101320 for 20 minutes at 4 °C). Primary conjugated (Table S1) were incubated for 15 minutes at 4 °C. Samples were acquired with Attune NxT (Invitrogen) flow cytometer and data were analyzed using FlowJo™ v10 Software (ThreeStar).

## Light transmission aggregometry (LTA)

TC-induced platelet aggregation (TCIPA) was analyzed by Light Transmission Aggregometry (LTA), equivalent for human and mouse. Aggregations were initiated in aggregation vials adding $600 \times 10^5/$200 μL of washed platelets or non-diluted cPRP. Next, 50 μL of TCs (or control beads) in TA buffer or serum-free culture media were added at up to $3 \times 10^6$/mL final concentration. Platelets' suspensions were stirred at 1100 rpm. Control samples were activated with 5 μM ADP in presence or absence of human fibrinogen (320 μg/mL, in-house generated[71]) or native equine tendon collagen Type 1 (2.5 μg/mL, Collagen Reagens HORM® Suspension, Takeda) in 300 μL final volume.

Prior to aggregation, TCs were washed and gently detached by non-enzymatic means (EDTA 0.48 mM, Versene 1×, Gibco), washed twice in RPMI-1640 or DMEM medium without serum + 0.1% BSA + 2 mM $CaCl_2$ + 1 mM $MgCl_2$, or in TA buffer. Platelets' aggregation was measured at 37 °C by a standard turbidimetric method in an APACT 4004 aggregometer (ELITech Group, Puteaux, France). Serum-free media or TA buffer were used as control together with PEG-treated 10 μm polystyrene beads (Phosphorex). Amplitudes of the aggregation curves generated with APACT LPC software were compared. The initial drop in light scattering due TCs addition to running aggregation vials was subtracted in Microsoft Excel, providing adjusted and comparable aggregation curves.

## Scanning electron microscopy

Gently detached TCs (EDTA 0.48 mM, Versene, Gibco), pre-washed in serum-free media (for cPRP) or TA buffer (for washed platelets), were incubated with platelet suspensions at the previously indicated concentrations. PEG (Polyethylene glycol) blocked beads were used as a control. LTA assay-derived TCs and platelet mixes were fixed 1:1 v/v in 2× fixative solution (4% paraformaldehyde, 5% glutaraldehyde in standard 0.1 M Sodium Cacodylate buffer, NaCac) and sedimented O/N at 4 °C. Then, pellets only were gently resuspended in NaCac Buffer and seeded on poly-L-lysine pre-coated (Sigma) glass coverslips in a 24-wells plate for 30 minutes at RT. Supernatants were removed, and coverslips were fully dried in an oven at 60 °C for 1 h. Following this, coverslips were washed with 0.2 μm-filtered NaCac buffer and $dH_2O$. Next, they were dehydrated with an increasing ethanol series (1 × 70%, 1 × 80%, 1 × 95% 5 min each and 2 × 100% 30 minutes each) and a 1,1,1,3,3,3-hexamethyldizilazane (HDMS, Merck, Millipore) graded series (25%, 50%, 75% and 100%, 5 min each). Cells were mounted onto 12 mm EM Aluminum Mounts for AMRAY (EMS, Hatfield) employing Leit-C conductive carbon cement (CCC, Plano GmbH, Germany). Samples were platinum metallized under vacuum using a Cressington Sputter Coater 208HR coupled to a Pfeiffer Vacuum (Germany). Image acquisition was performed at high resolution (10–15 kV) and ×6000–10,000 magnification on a PhenomWorld SEM desktop microscope (Phenom-World B.V, The Netherlands).

## CLEM sample preparation and image acquisition

Correlative light and electron microscopy (CLEM) was performed as previously described[72]. Briefly, ZF embryos were chemically fixed with 2.5% glutaraldehyde and 2% paraformaldehyde in 0.1 M NaCac buffer (pH 7.4) for 48 hr at 4 °C, then 3× washed with 0.1 M NaCac buffer, pH 7.4, and fixed again with 0,05% malachite green, 2.5% glutaraldehyde in 0.1 M NaCac buffer, pH 7.4 for 25 min in ice bath. Samples were postfixed for 45 min in 1% $OsO_4$ in 0.1 M NaCac buffer, pH 7.4 in a fume hood in ice bath then 2× washed in 0.1 M NaCac buffer (ice-cold to avoid thermal changes). Next, specimens were incubated in 1% aqueous tannic acid solution for 25 min in ice bath and 5× washed with distilled water (DW). Dehydration was performed with an ethanol's series (25%, 50%, 70%, 95%, and 100%) and acetone-dry at RT for 10 min. Then samples underwent a series of resin-acetone solution bath (1:3; 1:1; 3:1) finishing in 100% Epon resin for 3 × 1h at RT. Polymerization was performed in an oven at 60 °C for 48 h. The resin block was trimmed by ultramicrotomy. After targeting region of interest (ROI), 90 nm thin sections were collected and placed in slot Formvar cupper grids. The TEM data set was acquired with a Hitachi 7500 TEM, with 80 kV beam voltage, and the 8-bit images were obtained with a Hamamatsu camera C4742-51-12NR. The images were segmented using an open-source software ImageJ; AMIRA and IMARIS as follows: a) single images were stitched in a bigger composed image and combined into a 3D-stack; b) The combined 3D-stack was aligned and the contrast adjusted, by ImageJ. C) The segmentation of objects of interest, (arrested TCs, platelets) and 3D Volume reconstruction, by AMIRA and IMARIS.

## Tissue histology and immunohistochemistry

Human melanoma samples from lung metastases were obtained, processed, and stained as previously described[75]. Lungs harvested from mice sacrificed under isoflurane anesthesia and lidocaine analgesia (5 mg/kg) were washed and fixed O/N in 4% paraformaldehyde/phosphate-buffered saline (PBS) solution (EMS). Left lobes were treated for paraffine inclusion, while right ones for Optimal Cutting Temperature (OCT) inclusion (Tissue freezing media, Leica). Prior to paraffine embedding, O/N fixed lungs were 2× PBS, 2× ethanol 50% washed for 45 minutes then dehydrated O/N in 70% Ethanol. Tissues were processed in an automated Leica inclusion machine: 2 × 70%, 1 × 80%, 1 × 95% and 2 × 100% for 1 hour each, 2 × 45 min xylene/toluene each, and paraffin-embedded for 2 hours. Cryosections of 20 μm (for single cell analyses and segmentation) or 10 μm (for regular IF analyses) paraffin sections were mounted on poly-lysine-coated slides (SuperFrost UltraPlus, Thermo Fisher Scientific). Following a reverse dehydration scale, tissue antigens in paraffin-embedded specimens were retrieved by boiling in sodium citrate solution (10 mM, pH 6.0, Sigma). Unspecific antibody binding was blocked for 1 h in blocking solution (3% bovine serum albumin-BSA, 20 mM $MgCl_2$, 0.3% Tween 20, 5% FBS in PBS). Background and nonspecific staining controls were used. Primary antibodies (Table S1) were incubated O/N followed by secondary antibodies (Table S1) for 1 h room temperature prior to nuclei staining (DAPI-Thermo Fisher Scientific) and slides mounting (FluoromountG-SouthernBiotech). For biotinylated antibodies, tissue endogenous biotin was quenched with Avidin/Biotin blocking solution (Vector Laboratories) before secondary antibody. Antibody signal was developed after 1 h incubation with the Avidin/Biotin-HRP complex (Elite Vectastain ABComplex Kit, Vector Laboratories) and 1 h incubation with Streptavidin-647 (BioLegend).

## FFPE RNA tissue extraction and bulk RNA sequencing

Total RNA was isolated from 10 × 10 μm thin slides of paraffine-embedded lungs stocked at room temperature. NucleoSpin total RNA FFPE XS Micro kit (Cat. 740969.50, Macherey-Nagel) was used according to manufacturer's instruction, and RNA was eluted in 30 μL final volume. RNA concentration was assessed after extraction at the NanoPhotometer® N60 (Implen). RNA integrity (>50% DV200) was assessed by Bioanalyzer (total RNA Pico Kit and RNA 6000 Nano Kit, 2100 Instrument, Agilent Technologies, Paolo Alto, CA, USA). Sequencing libraries were prepared using "NEBNext Ultra II Directional RNA Library Prep Kit for Illumina "combined with "NEBNext rRNA Depletion Kit v2" for ribosomal RNA depletion (New England Biolabs, Ipswich, MA). Libraries were pooled and sequenced (paired-end, 2 × 100bp) on a NextSeq2000 according to the manufacturer's instructions (Illumina Inc., San Diego, CA, USA).

## Analysis of RNA-sequence reads: Identification of differentially expressed genes

For each sample, quality control was carried out and assessed with the NGS Core Tools FastQC[76] and QualiMap[77]. Sequence reads (minimum 88 million per sample) were mapped to *Mus musculus* mm10 using STAR[78] to obtain a BAM (Binary Alignment Map) file. An abundance matrix was generated after quantification with salmon[79] using default parameters. At last, differential expression analyses were performed using the DEseq2[80] package of the Bioconductor framework for RNA-Seq data[81]. Volcano plots were constructed based on $\log_2$ fold change (≥ or ≤0.5) and $-\log_{10}$ adjusted $p$ value (<1). Enrichment analysis of Gene Ontology (GO) terms was conducted using Metascape

(http://metascape.org)[82]. Gene selectively associated with immune cells were curated using ImmGen (https://www.immgen.org).

### Laser scanning confocal microscopy (LSCP)

For immunohistochemical analyses, images were acquired using a Leica TCS SP8 laser scan confocal microscope using a ×20/1.0 W-Plan-Apochromat objective (oil (NA 0.75) and a ×63 Plan-Apochromat [oil (NA 1.4)] magnification. For 3D segmentation of cells and platelet aggregates, no less than 10 z-stacks (1 AU) of single cells or clusters were acquired in 12-bit color depth and non-saturating settings.

For in vivo zebrafish imaging of TCs circulation and arrest, a Leica TCS SP8 microscope equipped using an M205 FA stereomicroscope (Leica) and a ×20/1.0 W-Plan-Apochromat objective (oil (NA 0.75)) on an SP8 confocal microscope (Leica) were used.

### Image processing and analysis

For metastatic foci number and size, macroscopic 8-bit lung (B16F10-seeded) images (MacOS 15.4, iPhone8) with identical zooming and size (100 × 100 px) were automatically thresholded on ImageJ. Binary masks obtained were analyzed by the "analyze particles" plugin with circularity set between 0.2–1.0 to exclude lung edge shadows and pixel size below 15 px considered as noise. Platelet aggregates and cell volumes of TCs intravascularly arrested in mice lungs were obtained after segmentation in AMIRA Visage 6 from the confocal acquisition of Z-stacks covering the whole cell-aggregate volume. For immuno-fluorescence images, brightness and contrast were equally adjusted in control and treated samples on ImageJ and/or Adobe Photoshop CS4 software.

### Statistics

Statistical analysis was performed with GraphPad Prism 9.4. The normal distribution of the data sets was assessed by the Shapiro–Wilk normality test. According to the number of data sets compared, the Mann–Whitney test or Kruskal–Wallis test followed by the original FDR method of Benjamini and Hochberg post-test were applied. In kinetics experiments, a two-way ANOVA followed by the original FDR method of Benjamini and Hochberg post test was used ($*p < 0.05$; $**p < 0.01$, $***p < 0.001$, $****p < 0.0001$). In all cases, the α-level was set at 0.05. All the data in graphs were presented as mean ± Standard Deviation (SD), except for the kinetics where mean ± SEM was used.

### Ethics statement

Human studies were performed according to the Helsinki declaration and regulations provided by the Ethics Committee from Hospital Gregorio Maranon, Madrid, Spain, Protocol Version 05/Junio 2018. Control human blood samples were obtained from volunteer blood donors who gave written informed consent recruited by the blood transfusion center where the research was performed (Etablissement Français du Sang, Grand Est). All procedures were registered and approved by the French Ministry of Higher Education and Research. The donors gave their approval in the CODHECO number AC- 2008 - 562 consent form for the samples to be used for research purposes. Immunocompetent and genetically modified mice used in this study were housed under pathogen-free conditions, and all procedures were performed in accordance with the European Union Guideline 2010/63/EU. The study was approved by the Regional Ethical Committee for Animal Experimentation of Strasbourg, CREMEAS (CEEA 35) under the APAFIS authorization 14741-2018041816337540 and 37433-2022052016445806. According to the ethical committee and approved experimental protocol, the signal of luminescence relative to D0 was considered as an indicator of tumor burden. The limit was set to a signal not superior at 10,000 times the input signal in the lungs. This value was never overcome in our experimental setting. Other parameters (Grimance pain scale and signs of suffering) were daily evaluated to ensure the well-being of the animals. Animals in conditions of suffering or presenting an excessive tumor burden were euthanized according to the scoring scale of the ethical approval protocol.

### Reporting summary

Further information on research design is available in the Nature Portfolio Reporting Summary linked to this article.

## Data availability

FFPE RNA sequencing data are publicly available in ArrayExpress accession E-MTAB-13864 at the following address: "www.ebi.ac.uk/biostudies/arrayexpress/studies/E-MTAB-13864". All the other data are available within the article and its Supplementary Information. Source data are provided with this paper.

## Code availability

MatLab scripts used for the heat mapping analysis in zebrafish (supplementary Fig. 2) are available here[2].

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

## Acknowledgements

We thank all members of JGG's team (past and present) for their constant discussions and help on this particular topic. J.G.G. is the coordinator of the NANOTUMOR Consortium, a program from ITMO Cancer of AVIESAN (Alliance Nationale pour les Sciences de la Vie et de la Santé, National Alliance for Life Sciences & Health) within the framework of the Cancer Plan (France). Work and people in the lab of J.G.G. are mostly supported by the INCa (Institut National Du Cancer, French National Cancer Institute), charities (La Ligue contre le Cancer, ARC (Association pour la Recherche contre le Cancer), FRM (Fondation pour la Recherche Médicale), the National Plan Cancer initiative, the Region Grand Est, INSERM and the University of Strasbourg. This work has been directly funded by the INCa grant (PLBIO 2016-164) and by the support of the Ligue contre le Cancer (labelisation), the Association Ruban Rose, and the SATT Conectus (Strasbourg). M.G.L. has been funded by the University of Strasbourg (IdeX, investissements d'Avenir), the INCa and the SATT Conectus (Strasbourg). M.P. and C.L. are supported by INCa, F.C. by NANOTUMOR, and V.M. by a Ph.D. fellowship from the French Ministry of Science (MESRI) and by the Foundation ARC (Association pour la recherche contre le Cancer grant number ARCDOC42023010006005). G.F. was supported by La Ligue contre le Cancer. L.B. is supported by FRM (Fondation pour la Recherche Médicale - ECO202206015567). We acknowledge PIC-STRA (imaging platform of the CRBS) for the support in image acquisition. We are grateful to Acticor for providing the Glenzocimab antibody and to François Bertucci for providing human samples. We are also thankful to recent donators (Rohan Athlétisme Saverne, Traileurs de la Rose) to support our work. Parts of the figure were drawn by using pictures from Servier Medical Art. Servier Medical Art by Servier is licensed under a Creative Commons Attribution 3.0 Unported License (https://creativecommons.org/licenses/by/3.0/).

## Author contributions

M.G.L. designed and conceived the project and performed most of the experiments, analyzed the data and wrote the manuscript; C.L. performed part of the mouse experiments, analyzed the data, wrote the manuscript and provided technical support during the revision; V.M. performed part of the mouse experiments, in vitro platelets-TCs interaction experiments, RNASeq data analysis, analyzed the data, wrote the manuscript and provided technical support during the revision process; L.B. performed lung immunophenotyping experiments, analyzed the data, and provided technical support during the revision process; G.F. performed zebrafish experiments and analyzed the data; C.M. performed platelets experiments and analyzed the data; I.B. performed CLEM experiments in zebrafish and analyzed the data; A.L. performed FFPE RNA extraction and provided technical support through all the project; F.C. provided technical support, revised the manuscript and prepared the figures; M.P. provided technical support with the zebrafish experiments; N.O. provided technical support during the revision and designed platelets-TCs interaction experiments; V.G. provided technical support with the zebrafish experiments; C.B. provided technical support; R.F. provided and stained human melanoma samples; A.P., N.P., A.M., R.C. performed RNASeq experiments, generated the libraries and analyzed the data; M.J.P. provided critical discussion through the project; O.L. contributed to design the project, performed part of the mouse experiments, provided technical support analyzed the data and wrote the manuscript; P.H.M. and J.G.G. designed, conceived and supervised the project, provided financial support and wrote the manuscript.

## Competing interests

M.P.J. is a scientific co-founder of Acticor Biotech and a scientific adviser. All other authors declare no competing interests.
