## [Peer Review File · Nature Communications]

Platelets favor the outgrowth of established metastasesThis manuscript has been previously reviewed at another journal that is not operating a transparent peer review scheme. This document only contains reviewer comments and rebuttal letters for versions considered at *Nature Communications*.

REVIEWERS' COMMENTS

Reviewer #4 (Remarks to the Author):

The original version of this Ms was reviewed by three referees (I was not one of them). I have reviewed the referee comments and the authors' responses and the final manuscript. The original reviews were positive but raised a significant number of questions and suggestions. The authors have responded at length with well reasoned answers and multiple revisions including new experiments and figures as well as significant modifications in the text - all reflecting highly responsive modification to address the referee comments and leading to an improved version of what was already a pretty good paper. This is now a very impressive piece of research extending and clarifying a long history of earlier publications on the topic of platelet effects on metastasis. The final version establishes multiple advances; effects of platelets, both early in seeding and later in outgrowth of metastases, immune suppression by platelets of established metastases and inhibition of the platelet enhancement of metastases (both early and late) by an antibody against the platelet receptor GPVI without any associated hemorrhage, thus opening the way to therapeutic inhibition of established metastases - a very important advance. This is a prime example of the value of effective peer review, including constructive review and a very effective response by the authors.

I have only one minor point; at a couple of points the authors use the word "impinge" where they mean "imply" or "implicate"
The paper is otherwise well written.

Reviewer #5 (Remarks to the Author):

The authors have thoroughly responded to Reviewer #2's comments and suggestions with a large number of new experiments and analysis, and strengthen the mechanistic insights (especially regulation of the immune microenvironment) of the manuscript. I think the manuscript is acceptable for publication at Nature Communications.

Reviewer #6 (Remarks to the Author):

The authors have sufficiently replied the concerns of reviewer 3.
From that no issues remain open.

The additional experiments for the revision process provide a detailed and impressive molecular mechanistic insight into the role of platelets in metastasis outgrowth, as such leading to a significant progress in the quality of the paper and the conclusions drawn. The novel data are sound and convincing in methodology.

The revised version of the paper, the spectrum of experimental approaches, the novelty of findings and in-depth interpretation justify a recommendation to accept this paper for publication in this revised form.

RESPONSE TO REVIEWERS' COMMENTS

First of all, we would like to thank the reviewers for their very positive comments and validation of the additional work we have provided, and thus judging that the paper is acceptable for publication. We provide here a final **point-by-point response** to the comments the reviewers had raised. **Our responses appear in green** in this document.

Reviewer #4 (Replacement for original R#1 from NCancer)

The original version of this Ms was reviewed by three referees (I was not one of them). I have reviewed the referee comments and the authors' responses and the final manuscript. The original reviews were positive but raised a significant number of questions and suggestions. The authors have responded at length with well-reasoned answers and multiple revisions including new experiments and figures as well as significant modifications in the text - all reflecting highly responsive modification to address the referee comments and leading to an improved version of what was already a pretty good paper. This is now a very impressive piece of research extending and clarifying a long history of earlier publications on the topic of platelet effects on metastasis. The final version establishes multiple advances; effects of platelets, both early in seeding and later in outgrowth of metastases, immune suppression by platelets of established metastases and inhibition of the platelet enhancement of metastases (both early and late) by an antibody against the platelet receptor GPVI without any associated hemorrhage, thus opening the way to therapeutic inhibition of established metastases - a very important advance. This is a prime example of the value of effective peer review, including constructive review and a very effective response by the authors. I have only one minor point; at a couple of points the authors use the word "impinge" where they mean "imply" or "implicate". The paper is otherwise well written.

Our answer: We warmly thank reviewer #4 for her/his favorable feedback and for acknowledging our efforts to provide a significantly improved version of our manuscript, and to address all the points initially raised by reviewer #1 to the best of our ability. We have now replaced the word 'impinge' in the text.

Reviewer #5: (Replacement for original R#2 from NCancer)

The authors have thoroughly responded to Reviewer #2' comments and suggestions with a large number of new experiments and analysis, and strengthen the mechanistic insights (especially regulation of the immune microenvironment) of the manuscript. I think the manuscript is acceptable for publication at Nature Communications.

Our answer: We also express our gratitude to reviewer #5 for her/his favorable feedback and clear support towards the publication of our work in Nature Communications.

Reviewer #6: (Replacement for original R#3 from NCancer)

The authors have sufficiently replied the concerns of reviewer 3. From that no issues remain open. The additional experiments for the revision process provide a detailed and impressive molecular mechanistic insight into the role of platelets in metastasis outgrowth, as such leading to a significant progress in the quality of the paper and the conclusions drawn. The novel data are sound and convincing in methodology. The revised version of the paper, the spectrum of experimental approaches, the novelty of findings and in-depth interpretation justify a recommendation to accept this paper for publication in this revised form.

Our answer: We extend our sincere gratitude to reviewer #4 for her/his positive feedback, acknowledging our dedication to delivering a substantially enhanced manuscript. Additionally, we appreciate reviewer #4's recognition of our commitment to addressing the

concerns initially raised by reviewer #6 fully, and recommendation for publication in Nature Communication.